# Is Your LiDAR Placement Optimized for 3D Scene Understanding?

Ye Li[1]    Lingdong Kong[2]    Hanjiang Hu[3]    Xiaohao Xu[1]    Xiaonan Huang[1]

[1]University of Michigan, Ann Arbor    [2]National University of Singapore
[3]Carnegie Mellon University

https://github.com/ywyeli/Place3D

## Abstract

The reliability of driving perception systems under unprecedented conditions is crucial for practical usage. Latest advancements have prompted increasing interest in multi-LiDAR perception. However, prevailing driving datasets predominantly utilize single-LiDAR systems and collect data devoid of adverse conditions, failing to capture the complexities of real-world environments accurately. Addressing these gaps, we proposed **Place3D**, a full-cycle pipeline that encompasses Li-DAR placement optimization, data generation, and downstream evaluations. Our framework makes three appealing contributions. **1)** To identify the most effective configurations for multi-LiDAR systems, we introduce the Surrogate Metric of the Semantic Occupancy Grids (M-SOG) to evaluate LiDAR placement quality. **2)** Leveraging the M-SOG metric, we propose a novel optimization strategy to refine multi-LiDAR placements. **3)** Centered around the theme of multi-condition multi-LiDAR perception, we collect a 280,000-frame dataset from both clean and adverse conditions. Extensive experiments demonstrate that LiDAR placements optimized using our approach outperform various baselines. We showcase exceptional results in both LiDAR semantic segmentation and 3D object detection tasks, under diverse weather and sensor failure conditions.

## 1 Introduction

Accurate 3D perception plays a crucial role in autonomous driving, involving detecting the objects around the vehicle and segmenting the scene into meaningful semantic categories. LiDARs are becoming crucial for driving perception due to their capability to capture detailed geometric information about the surroundings [3, 109, 53, 54, 118]. While the latest models achieved promising accuracy on standard datasets, *e.g.*, nuScenes [7] and SemanticKITTI [6], improving the resilience of perception under corruptions and sensor failures is still a critical yet under-explored task [46, 94, 5, 95].

Recent studies have primarily focused on refining the sensing systems by designing new algorithms with novel model architectures [15] or 3D representations [28, 99, 11]. The selection of LiDAR configurations, however, often relies on industry experience and design aesthetics. Therefore, existing literature has potentially overlooked optimal LiDAR placements for maximum sensing efficacy.

An intuitive approach for optimizing LiDAR configurations involves a comprehensive cycle of data collection, model training, and validation across various LiDAR setups to enhance the autonomous driving system's perception accuracy [24, 52]. However, this approach faces significant challenges due to the substantial computational resources and extensive time required for data collection and processing [57, 17]. Although recent works [34, 65] made preliminary attempts to explore the impact

38th Conference on Neural Information Processing Systems (NeurIPS 2024).

of LiDAR placements on perception accuracy, they have neither proposed an optimization method nor evaluated the performance in adverse conditions.

In this work, we delve into the optimization of sensor configurations for autonomous vehicles by tackling two critical sub-problems: **1)** the performance evaluation of sensor-specific configurations, and **2)** the optimization of these configurations for enhanced 3D perception tasks, encompassing 3D object detection and LiDAR semantic segmentation. To achieve this goal, we propose a systematic LiDAR placements evaluation and optimization framework, dubbed **Place3D**. The overall pipeline is endowed with the capability to synthesize point cloud data with customizable configurations and diverse conditions, including common corruptions, external disturbances, and sensor failures.

We first introduce an easy-to-compute Surrogate Metric of Semantic Occupancy Grids (M-SOG) to evaluate the equality of LiDAR placements. Next, we propose a novel optimization approach utilizing our surrogate metric based on Covariance Matrix Adaptation Evolution Strategy (CMA-ES) to find the near-optimal LiDAR placements. To verify the correlation between our surrogate metric and assess the effectiveness of our optimization approach on both clean and adverse conditions, we design an automated multi-condition multi-LiDAR data simulation platform and establish a comprehensive benchmark consisting of a total of seven LiDAR placement baselines inspired by existing self-driving configuration from the autonomous vehicle companies.

Our benchmark, along with a large-scale multi-condition multi-LiDAR perception dataset, encompasses state-of-the-art learning-based perception models for 3D object detection [51, 107, 63, 109] and LiDAR semantic segmentation [18, 111, 86, 117, 50], as well as six distinct adverse conditions coped with weather and sensor failures [46, 95]. Utilizing the proposed framework, we explored how various perturbations and downstream 3D perception tasks affect optimization outcomes.

To summarize, this work makes the following key contributions:

- To the best of our knowledge, **Place3D** serves as the first attempt at investigating the impact of multi-LiDAR placements for 3D semantic scene understanding in diverse conditions.
- We introduce M-SOG, an innovative surrogate metric to effectively evaluate the quality of LiDAR placements for both detection (sparse 3D) and segmentation (dense 3D) tasks.
- We propose a novel optimization approach utilizing our surrogate metric to refine LiDAR placements, which exhibit excellent LiDAR semantic segmentation and 3D object detection accuracy and robustness, outperforming baselines for relatively 9% in metrics.
- We contribute a 280,000-frame multi-condition multi-LiDAR point cloud dataset and establish a comprehensive benchmark for LiDAR-based 3D scene understanding evaluations. We hope this work can lay a solid foundation for future research in this relevant field.

## 2 Related Work

**LiDAR Sensing.** LiDAR sensing is critical in autonomous vehicles, providing essential structural information for their operation [59, 3, 73]. Utilizing 3D data, LiDAR supports tasks such as semantic scene understanding [7, 26, 29, 6, 85, 48, 10], generation [75, 120, 74, 97], decision making [25, 8, 19, 87], and simulation [23, 68, 67, 93]. This work specifically targets LiDAR-based perception and simulation, which are at the forefront of current research in autonomous vehicle technology.

**LiDAR-Based 3D Scene Understanding.** LiDAR segmentation and 3D object detection are key tasks for 3D scene understanding. Segmentation models are categorized into point-based [88, 35, 36, 110, 77], range view [70, 90, 21, 98, 113, 16, 47, 2, 45, 99], bird's eye view [111, 115, 12], voxel-based [18, 86, 117, 33], and multi-view fusion [58, 38, 78, 60, 62, 14, 13, 61, 100] methods. While they show promising results on benchmarks, their performance across different LiDAR configurations is not well explored. For 3D object detection, models typically use point-based [82, 106, 105, 116] or voxel-based [114, 51, 69, 92, 83, 55, 66, 104] representations, with recent works favoring fusion for improved detection [64, 80, 81, 63, 56]. Our study complements these approaches by optimizing LiDAR placements to enhance performance under various real-world conditions.

**LiDAR Perception Robustness.** The reliability of LiDAR-based 3D scene understanding models in real-world scenarios is crucial [84, 39]. Recent studies have explored the robustness of 3D perception models against adversarial attacks [112, 96], common corruptions [46, 49, 94, 103, 43, 4], adverse weather [31, 30, 79, 37], sensor failures [108, 15, 28], and combined motion and sensor perturbations

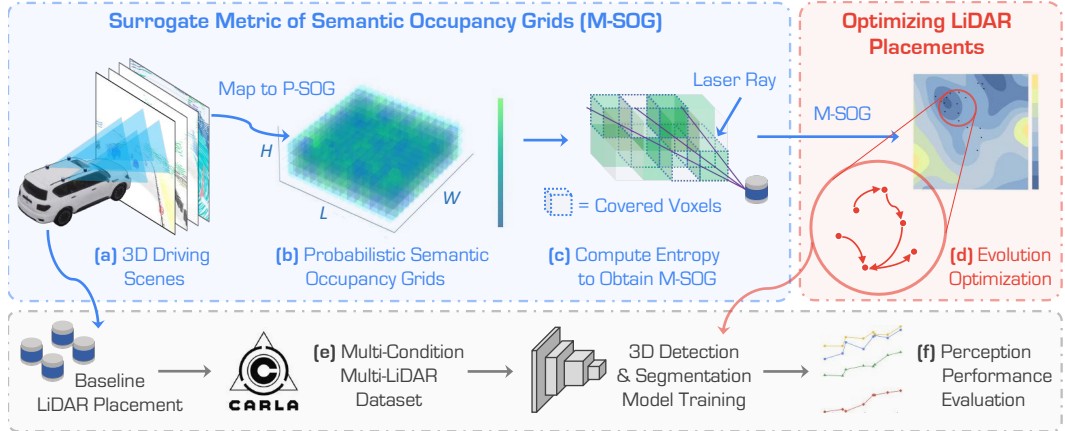

Figure 1: **Place3D pipeline for multi-LiDAR placement optimization.** We first utilize the semantic point cloud synthesized in CARLA (a) to generate Probabilistic SOG (b) and obtain voxels covered by LiDAR rays to compute M-SOG (c). We propose a CMA-ES-based optimization strategy to maximize M-SOG, finding optimal LiDAR placement (d). To verify the effectiveness of our LiDAR placement optimization strategy, we contribute a multi-condition multi-LiDAR dataset (e) and evaluate the performance of baselines and optimized placements on both clean and corruption data (f).

[102, 95]. To our knowledge, no prior work has focused on optimizing LiDAR placement to improve semantic 3D scene understanding robustness. Our Place3D addresses this gap by studying various placement strategies to enhance robustness in both in-domain and out-of-domain scenarios.

**Sensor Placement Optimization.** Optimizing sensor placements can enhance performance and address the challenges of heuristic design [42, 101]. In autonomous driving, optimizing LiDAR placements is relatively new [65]. Hu *et al.* [34] studied multi-LiDAR placements for 3D object detection, while Li *et al.* [57] examined LiDAR-camera configurations for multi-modal detection. Other works [41, 44, 9, 40] explored roadside LiDAR placements for V2X applications. Distinguishing from these efforts, our work is the first to investigate LiDAR placements for robust 3D scene understanding in challenging conditions. We establish benchmarks for both LiDAR semantic segmentation and 3D object detection, providing an in-depth analysis of optimal placements for robust perception.

## 3 Place3D: A Full-Cycle LiDAR Placement Pipeline

In this section, we introduce the Surrogate Metric of Semantic Occupancy Grids (M-SOG) to assess the 3D perception performance of sensor configurations (Section 3.2). Leveraging M-SOG, we propose a novel optimization strategy for multi-LiDAR placement (Section 3.3). Our approach is theoretically grounded; we provide an optimality certification to ensure that optimized placements are close to the global optimum (Section 3.4). Figure 1 depicts an overview of our Place3D framework.

### 3.1 Preliminary Formulation of Surrogate Metric

**Region of Interest.** Following the literature of sensor placement [34, 57], we define the Region of Interest (ROI) for perception as a finite 3D cuboid space $[l, w, h]$ with ego-vehicle in the center, and divide the ROI space $S$ into voxels with resolution $[\delta_l, \delta_w, \delta_h]$ as $S = \{v_1, v_2, ..., v_N\}$, where $N = \frac{l}{\delta_l} \times \frac{w}{\delta_w} \times \frac{h}{\delta_h}$ denotes the total number of voxels.

**Probabilistic Occupancy Grids (POG).** We define POG as the joint probability of all non-zero voxels in the ROI as $p_{POG} = p(v_1, v_2, ..., v_N) = \prod_{i=1, \, p(v_i) \neq 0}^{N} p(v_i)$, where each voxel is assumed to be independently and identically distributed. The voxel $v_i$ is occupied if it is within any 3D bounding box $b_c$ of object class $c$ at frame $t$, which is denoted as $v_i = b_c^{(t)}$. Therefore, the probability of being occupied for voxel $v_i$ among all $T$ frames can be estimated as $p(v_i) = \sum_{t=1}^{T} \mathbb{1}(v_i = b_c^{(t)})/T$.

**Probabilistic Surrogate Metric.** Now given some LiDAR placement $L$, the conditional POG can be found as the conditional joint distribution $p_{POG|L} = p(v_1, v_2, ..., v_N | L)$ in the literature [34].

To show how much uncertainty the placement $L$ can reduce in the ROI, the naive surrogate metric *S-MIG* [34] is introduced as the reduction of the entropy as $-H_{POG|L} = -\sum_{i=1}^{N} H(v_i|L)$ with constant $H_{POG} = \sum_{i=1}^{N} H(v_i)$, where $H(v_i), H(v_i|L)$ can be found through the entropy of Bernoulli distribution $p(v_i), p(v_i|L)$, respectively.

## 3.2 M-SOG: Surrogate Metric of Semantic Occupancy Grids

The naive surrogate metric *S-MIG* [34] employs 3D bounding boxes as priors to understand 3D scene distribution. However, this approach exhibits substantial deviations from the actual physical boundaries of the objects and overlooks the occlusion relationships between objects and the environment in LiDAR applications. Furthermore, it is limited to detection tasks only. To overcome these limitations, we propose the Surrogate Metric of Semantic Occupancy Grids to assess and optimize LiDAR placement for 3D scene understanding tasks. Leveraging the Semantic Occupancy Grids derived from diverse environments, our approach can be expanded to tackling adverse conditions.

**Semantic Occupancy Grids (SOG).** Given a 3D driving scene with a set of semantic classes $Y = \{y_1, y_2, ..., y_M\}$, where $M$ represents the total number of semantic classes in the scene and assuming that each voxel can only be occupied by one semantic tag for each frame, we denote $y^{(t)}(v_i)$ as the semantic class of voxel $v_i \in \{v_1, v_2, ..., v_N\}$ at time frame $t \in \{1, 2, ..., T\}$. Notably, empty voxels are also considered a semantic class. Accordingly, the set of voxels occupied by semantic class $y_c$ at frame $t$ is defined as $S_{y_c}^{(t)} = \{v_i | y^{(t)}(v_i) = y_c\}$, $c = 1, 2, ..., M$, $t = 1, 2, ..., T$. Then, we introduce SOG to describe the total semantic voxel distribution:

$$S_{SOG}^{(t)} = \{S_{y_1}^{(t)}, S_{y_2}^{(t)}, ..., S_{y_M}^{(t)}\}, \ t = 1, 2, ..., T . \tag{1}$$

**Probabilistic Semantic Occupancy Grids.** In contrast to the Bernoulli distribution *POG* [34], we propose a more accurate Multinomial distribution: Probabilistic Semantic Occupancy Grids (*P-SOG*), denoted as $p_{SOG}$, to represent the probability distribution of voxels belonging to certain semantic classes. Before estimating $p_{SOG}$, we first traverse all frames from given scenes to obtain the probability $\hat{p}$ for each voxel $v_i$ occupied by each semantic class $y_c$:

$$\hat{p}(v_i = y_c) = \frac{|\{t \in \{1, 2, ..., T\}|v_i \in S_{y_c}^{(t)}\}|}{T}, \ c = 1, 2, ..., M . \tag{2}$$

Notably, we denote voxel $v_i$ being occupied by semantic class $y_c$ as $v_i = y_c$, and $\hat{p}$ indicates estimated distributions from observed samples, whereas $p$ refers to the statistical parameters to be estimated, which are unknown and non-random. We compute the joint probability of all non-zero voxels in the ROI to estimate the $p_{SOG}$. Following the literature [34], the joint distribution over the voxel set $S$ is:

$$\hat{p}_{SOG} = \hat{p}(v_1, v_2, ..., v_N) = \prod_{i=1, \ \hat{p}(v_i) \neq 0}^{N} \hat{p}(v_i) . \tag{3}$$

The uncertainty of Probabilistic Semantic Occupancy Grids reflects the 3D scene understanding capability of sensors within a given scene. To quantify this uncertainty, we calculate the entropy for the probability distribution of each voxel $v_i$ in $\hat{p}_{SOG}$ as: $\hat{H}(v_i) = -\sum_{c=1}^{M} \hat{p}(v_i = y_c) \log \hat{p}(v_i = y_c)$.

Based on the independent and identically distributed assumption of *SOG*, the entropy of *P-SOG* over the voxel set $S$ is given by: $\hat{H}_{SOG} = \mathbb{E}_{v_i \sim p_S} \sum_{i=1}^{N} \hat{H}(v_i)$, where $p_S$ denotes the $p_{SOG}$ over the voxel set $S$. From the perspective of density estimation, the true *P-SOG* and its corresponding entropy can be estimated as $p_{SOG} = \hat{p}_{SOG}$ and $H_{SOG} = \hat{H}_{SOG}$, respectively.

**Surrogate Metric of Semantic Occupancy Grids.** To evaluate LiDAR placements, we further analyze the joint probability distribution of voxels covered by LiDAR rays with varied LiDAR placements. Leveraging the Amanatides and Woo's Fast Voxel Traversal Algorithm [1], we obtain the voxels covered by LiDAR rays as $S|L_j = \text{FastVT}(S, L_j) = \{v_1^{L_j}, v_2^{L_j}, ..., v_{N_j}^{L_j}\}$ given LiDAR configuration $L = L_j, j = 1, 2, ..., J$, where $j$ indexes the LiDAR placements, $J$ is the number of total configurations, and $N_j$ is the number of voxels covered by rays of LiDAR configuration $L = L_j$. Then, the semantic entropy distribution of *P-SOG* over the voxel set $S|L_j$ can be estimated as:

$$H_{SOG}^{L_j} = \mathbb{E}_{v_i^{L_j} \sim p_{S|L_j}} \sum_{i=1}^{N_j} \hat{H}(v_i^{L_j}) . \tag{4}$$

**Algorithm 1** Multi-LiDAR Placement Optimization for Semantic 3D Scene Understanding

1: Initialize: $k \leftarrow 0$, $\mathbf{m}^{(0)}$, $\sigma^{(0)}$, $\mathbf{C}^{(0)}$, $N_k$, $M_k$, $\forall k \in \{0, 1, 2, \ldots, K\}$
2: **for** $k = 0, 1, 2, \ldots, K$ **do**
3:     **for** $i = 1$ to $N_k$ **do**
4:         Sample $\mathbf{u}_i^{(k)} \sim \mathcal{N}(\mathbf{m}^{(k)}, (\sigma^{(k)})^2 \mathbf{C}^{(k)})$ from $\delta$-density gird-level candidates
5:         Calculate $G(\mathbf{u}_i^{(k)})$
6:     **end for**
7:     Update $\mathbf{m}^{(k+1)}$ based on the top $M_k$ best solutions $\hat{\mathbf{u}}_i^{(k)}$ via Equation (6)
8:     Update $\sigma^{(k+1)}$ and $\mathbf{C}^{(k+1)}$ via Equation (7), (8), (9), and (10)
9: **end for**

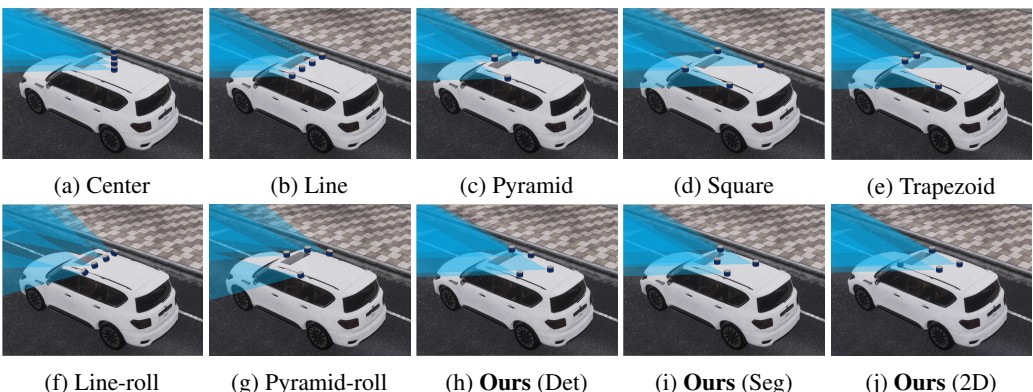

| (a) Center | (b) Line | (c) Pyramid | (d) Square | (e) Trapezoid |
| (f) Line-roll | (g) Pyramid-roll | (h) **Ours** (Det) | (i) **Ours** (Seg) | (j) **Ours** (2D) |

Figure 2: **Visualized LiDAR Placements.** We compare the LiDAR placements optimized from our proposed M-SOG metric (for LiDAR semantic segmentation and 3D object detection) and heuristic LiDAR placements utilized by major autonomous vehicle companies (see Appendix Section B.1).

We further define the information gain of 3D scene understanding as $\Delta H = H_{SOG} - H_{SOG}^{L_j}$ to describe the perception capability of LiDAR configuration $L_j$. Since $H_{SOG}$ is a constant given certain 3D semantic scenes with the fixed *P-SOG* distribution, we propose the normalized Surrogate Metric of Semantic Occupancy Grids (M-SOG) as follows to evaluate the perception capability:

$$M_{SOG}(L_j) = -\frac{1}{N_j} H_{SOG}^{L_j} = -\frac{1}{N_j} \mathbb{E}_{v_i^{L_j} \sim p_{S|L_j}} \sum_{i=1}^{N_j} H(v_i^{L_j})$$

$$= \frac{1}{N_j} \sum_{i=1}^{N_j} \sum_{c=1}^{M} \hat{p}(v_i^{L_j} = y_c) \log \hat{p}(v_i^{L_j} = y_c) . \qquad (5)$$

### 3.3 Sensor Configuration Optimization

We adopt the heuristic optimization based on the Covariance Matrix Adaptation Evolution Strategy (CMA-ES) [32] to find an optimized LiDAR configuration.

**Objective Function.** We define the objective as $F(\mathbf{u}_j) = M_{SOG}(L_j)$, where $\mathbf{u}_j \in \mathcal{U} \subset \mathbb{R}^n$ represents the LiDAR configuration $L_j$ and is subject to the physical constraint $P(\mathbf{u}_j) = 0$, and $P(\mathbf{u}) > 0$ if it is violated, *e.g.*, mutual distance between LiDARs and distance from a 2D plane. In our optimization, $\mathbf{u}_j$ includes the 3D coordinates and rolling angles of LiDARs, $\mathcal{U}$ is the finite cuboid space above the vehicle roof. Without ambiguity, we omit subscript $j$ in the following text. We then transform the constrained optimization into the unconstrained Lagrangian form as $G(\mathbf{u}) = -F(\mathbf{u}) + \lambda P(\mathbf{u})$, where $\lambda$ is the Lagrange multiplier. We optimize $G(\mathbf{u})$ through an iterative process that adapts the distribution of candidate solutions as Algorithm 1.

**Optimization Approach.** We first define a multivariate normal distribution $\mathcal{N}(\mathbf{m}^{(k)}, (\sigma^{(k)})^2 \mathbf{C}^{(k)})$, where $\mathbf{m}^{(k)}$, $\sigma^{(k)}$, and $\mathbf{C}^{(k)}$ are the mean vector, step size, and covariance matrix of the distribution of iteration $k$, respectively. We discretize the configuration space $\mathcal{U}$ with density $\delta$ and sample $N_k$

candidates $u_i^{(k)} \sim \mathcal{N}(\mathbf{m}^{(k)}, (\sigma^{(k)})^2 \mathbf{C}^{(k)})$ in each iteration $k$, where $i$ indexes the candidates. We update mean vector $\mathbf{m}^{k+1}$ for the next iteration $k+1$ as the updated center of the search distribution for LiDAR configuration. The overall process can be depicted as follows:

$$\mathbf{m}^{(k+1)} = \sum_{i=1}^{M_k} w_i \hat{\mathbf{u}}_i^{(k)}, \ G(\hat{\mathbf{u}}_1^{(k)}) \le G(\hat{\mathbf{u}}_2^{(k)}) \le \cdots \le G(\hat{\mathbf{u}}_{M_k}^{(k)}) , \tag{6}$$

where $M_k$ is the number of best solutions we adopt to generate $\mathbf{m}^{(k+1)}$, and $w_i$ are the weights based on solution fitness. We obtain the evolution path $\mathbf{p}_{\mathbf{C}}^{(k+1)}$ that accumulates information about the direction of successful steps as follows:

$$\mathbf{p}_{\mathbf{C}}^{(k+1)} = (1 - c_{\mathbf{C}}) \cdot \mathbf{p}_{\mathbf{C}}^{(k)} + \sqrt{1 - (1 - c_{\mathbf{C}})^2} \cdot \sqrt{\frac{1}{\sum_{i=1}^{M_k} w_i^2}} \cdot \frac{\mathbf{m}^{(k+1)} - \mathbf{m}^{(k)}}{\sigma^{(k)}} , \tag{7}$$

where $c_{\mathbf{C}}$ is the learning rate for the covariance matrix update. The covariance matrix $\mathbf{C}$ controls the shape and orientation of the search distribution for LiDAR configurations. It can be updated at each iteration $k$ in the following format:

$$\mathbf{C}^{(k+1)} = (1 - c_{\mathbf{C}}) \mathbf{C}^{(k)} + c_{\mathbf{C}} \mathbf{p}_{\mathbf{C}}^{(k+1)} \mathbf{p}_{\mathbf{C}}^{(k+1)^T} . \tag{8}$$

Similarly, we update $\mathbf{p}_\sigma$ as the evolution path for step size adaptation. Then, the global step size $\sigma$ can be found below for the scale of search to balance exploration and exploitation:

$$\mathbf{p}_\sigma^{(k+1)} = (1 - c_\sigma) \mathbf{p}_\sigma^{(k)} + \sqrt{1 - (1 - c_\sigma)^2} \cdot \sqrt{\frac{1}{\sum_{i=1}^{M_k} w_i^2}} \cdot \frac{\mathbf{m}^{(k+1)} - \mathbf{m}^{(k)}}{\sigma^{(k)}} , \tag{9}$$

$$\sigma^{(k+1)} = \sigma^{(k)} \exp\left( \frac{c_\sigma}{d_\sigma} \left( \frac{\|\mathbf{p}_\sigma^{(k+1)}\|}{E\|\mathcal{N}(0, \mathbf{I})\|} - 1 \right) \right) , \tag{10}$$

where $c_\sigma$ is the learning rate for updating the evolution path $\mathbf{p}_\sigma$. $d_\sigma$ is a normalization factor to calibrate the pace at which the global step size is adjusted.

## 3.4 Theoretical Analysis

Once the evolution optimization empirically converges, it holds that the optimized solution is the local optima of the $\delta$-density Grids space of LiDAR configuration space. In this section, we provide a stronger optimality certification to theoretically ensure that the optimized placement is close to the global optimum under the assumption of bounded and smooth objective function. Due to space limits, the full proof has been attached to the Appendix Section E.

**Theorem 1** (Optimality Certification). *Given the continuous objective function $G : \mathbb{R}^n \to \mathbb{R}$ with Lipschitz constant $k_G$ w.r.t. input $\boldsymbol{u} \in \mathcal{U} \subset \mathbb{R}^n$ under $\ell_2$ norm, suppose over a $\delta$-density Grids subset $S \subset \mathcal{U}$, the distance between the maximal and minimal of function $G$ over $S$ is upper-bounded by $C_M$, and the local optima is $\boldsymbol{u}_S^* = \arg\min_{\boldsymbol{u} \in S} G(x)$, the following optimality certification regarding $x \in \mathcal{U}$ holds that:*

$$\|G(\boldsymbol{u}^*) - G(\boldsymbol{u}_S^*)\|_2 \le C_M + k_G \delta , \tag{11}$$

*where $\boldsymbol{u}^*$ is the global optima over $\mathcal{U}$.*

The global optimality certification Theorem 1 is applicable in practice because the Lipschitz constant $k_G$ and the distance between the maximal and minimal of objective function $G$ over $S$ can be approximated easily through calculating $G(\mathbf{u}_i^{(k)})$ of Algorithm 1 for each sampled $\mathbf{u}_i^{(k)}$ over the $\delta$-density Grids subset. Besides, we have a more general corollary below to further relax the assumption of bounded objective function.

**Corollary 1.** *When $\mathcal{U}$ is a hyper-rectangle with the bounded $\ell_2$ norm of domain $U_i \in \mathbb{R}$ at each dimension $i$, with $i = 1, 2, \ldots, n$, Theorem 1 can hold in a more general way by only assuming that the Lipschitz constant $k_G$ of the objective function is given, where Equation (11) becomes:*

$$\|G(\boldsymbol{u}^*) - G(\boldsymbol{u}_S^*)\|_2 \le k_G \sum_{i=1}^{n} U_i + k_G \delta . \tag{12}$$

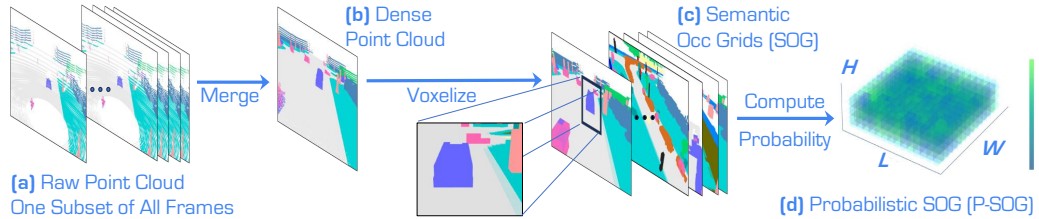

Figure 3: **Pipeline of Probabilistic SOG generation.** We first merge multiple frames of raw point clouds (a) into dense point clouds (b). Then, we voxelize dense point clouds into SOG, *i.e.*, semantic occupancy grids (c), and traverse all frames of dense point clouds to synthesize probabilistic SOG (d).

Table 1: **Optimized M-SOG Metrics.** We report the $M_{SOG}$ scores for both detection and segmentation tasks in our pipeline. The scores are calculated based on the *car* class for detection and all semantic classes for segmentation. *Line-roll* and *Pyramid-roll* are abbreviated as *L-roll* and *P-roll*.

| Metrics [$M_{SOG}$] | Center | Line | Pyramid | Square | Trapezoid | L-roll | P-roll | **Ours** |
|---|---|---|---|---|---|---|---|---|
| 3D Detection ($\times 10^{-6}$) | $-1.26$ | $-1.65$ | $-1.34$ | $-1.54$ | $-1.52$ | $-1.41$ | $-1.35$ | $-1.18$ |
| 3D Segmentation ($\times 10^{-6}$) | $-1.58$ | $-2.62$ | $-2.56$ | $-2.35$ | $-2.89$ | $-3.13$ | $-1.63$ | $-1.29$ |

## 4  Experiments

### 4.1  Benchmark Setups

**Data Generation.** We generate LiDAR point clouds and ground truth using CARLA [23]. We use the maps of Towns 1, 3, 4, and 6 and set 6 ego-vehicle routes for each map. We incorporate 23 semantic classes for LiDAR semantic segmentation and 3 instance classes for 3D object detection. Data collection is performed for 10 LiDAR placements, resulting in a total of 280,000 frames: **1)** For each placement, we gather 340 clean scenes and 360 corrupted scenes, with each scene consisting of 40 frames. **2)** The clean set comprises 13,600 frames, including 11,200 samples (280 scenes) for training and 2,400 samples (60 scenes) for validation, following the split ratio used in nuScenes [7]. **3)** The corruption set is used for robustness evaluation and contains 6 different conditions, each with 2,400 samples (60 scenes). More details are in the Appendix Section A.

**LiDAR Configurations.** We adopt five commonly employed heuristic LiDAR placements, which have been adopted by leading autonomous driving companies, as our baseline. These placements are represented in Figure 2a-e. Following KITTI [29], we configured the LiDAR sensor with a vertical field of view of [-24.8, 2.0] degrees. To achieve the *Line-roll* and *Pyramid-roll* configurations, we adjusted LiDAR roll angles for the *Line* and *Pyramid* setups, as depicted in Figure 2. We present detailed LiDAR configurations in the Appendix Section B.1.

**Corruption Types.** To replicate adverse conditions, we synthesized six types of corrupted point clouds on the validation set of each sub-set, following the settings of the Robo3D benchmark [46]. These corruptions can be categorized into 1) severe weather conditions, including "snow", "fog", and "wet ground", 2) external disturbances, including "motion blur", and 3) internal sensor failures, including "crosstalk" and "incomplete echo". We include the features and implementation details of corruptions in the Appendix Section B.2.

**P-SOG Synthesis.** We follow the steps as depicted in Figure 3 to generate semantic occupancy grids for each LiDAR scene. We first collect point clouds and semantic labels using high-resolution LiDARs and sequentially divide all samples into multiple subsets. Through the transformation of world coordinates of the ego-vehicle, frames of point clouds of each subset are aggregated into one frame of dense point cloud. Then, we utilize the voting strategy to determine the semantic label for each voxel in ROI and generate the SOG. This process is executed across all subsets to produce P-SOG. Notably, the P-SOG is only generated on the scenes of the training set.

**Surrogate Metric of SOG.** We compute the scores of M-SOG separately for 3D detection and segmentation, as shown in Table 1. To evaluate M-SOG for segmentation, we utilize all semantic classes to generate semantic occupancy grids. For detection, we specifically focus on the *car* semantic type, while merging the remaining semantic classes into a single category for M-SOG analysis.

Table 2: **Performance evaluations of LiDAR semantic segmentation** under clean and adverse conditions. For each LiDAR placement strategy, we report the mIoU scores (↑), represented in percentage (%). The average scores only include adverse scenarios.

| Method | Center | | | Line | | | Pyramid | | | Square | | |
|---|---|---|---|---|---|---|---|---|---|---|---|---|
| | Mink | SPV | Cy3D | Mink | SPV | Cy3D | Mink | SPV | Cy3D | Mink | SPV | Cy3D |
| Clean ● | 65.7 | 67.1 | 72.7 | 59.7 | 59.3 | 68.9 | 62.7 | 67.6 | 68.4 | 60.7 | 63.4 | 69.9 |
| Fog ○ | 55.9 | 39.3 | 55.6 | 51.7 | 42.8 | 55.5 | 52.9 | 48.6 | 51.0 | 55.6 | 40.7 | 52.0 |
| Wet Ground ○ | 63.8 | 66.6 | 64.4 | 60.2 | 57.9 | 66.4 | 60.3 | 66.6 | 52.2 | 61.9 | 64.3 | 55.6 |
| Snow ○ | 25.1 | 35.6 | 16.7 | 35.5 | 31.3 | 4.7 | 25.2 | 30.2 | 5.0 | 33.5 | 38.3 | 2.7 |
| Motion Blur ○ | 35.8 | 35.6 | 37.6 | 52.0 | 46.1 | 39.4 | 50.7 | 55.1 | 42.5 | 51.5 | 53.9 | 44.2 |
| Crosstalk ○ | 24.7 | 19.5 | 36.9 | 27.1 | 13.6 | 34.3 | 17.3 | 14.8 | 26.6 | 26.5 | 18.6 | 37.1 |
| Incomplete Echo ○ | 64.5 | 66.8 | 71.5 | 59.2 | 57.1 | 68.3 | 60.2 | 65.9 | 60.9 | 61.2 | 63.7 | 68.7 |
| Average ● | 45.0 | 43.9 | 47.1 | 47.6 | 41.5 | 44.8 | 44.4 | 46.9 | 39.7 | 48.4 | 46.6 | 43.4 |

| Method | Trapezoid | | | Line-Roll | | | Pyramid-Roll | | | **Ours** | | |
|---|---|---|---|---|---|---|---|---|---|---|---|---|
| | Mink | SPV | Cy3D | Mink | SPV | Cy3D | Mink | SPV | Cy3D | Mink | SPV | Cy3D |
| Clean ● | 59.0 | 61.0 | 68.5 | 58.5 | 60.6 | 69.8 | 62.2 | 67.9 | 69.3 | 66.5 | 68.3 | 73.0 |
| Fog ○ | 49.7 | 40.9 | 52.1 | 48.6 | 42.2 | 49.7 | 52.2 | 47.2 | 50.7 | 59.5 | 59.1 | 57.6 |
| Wet Ground ○ | 60.4 | 61.3 | 64.6 | 59.2 | 62.0 | 65.4 | 60.9 | 67.1 | 67.9 | 66.6 | 66.7 | 67.2 |
| Snow ○ | 27.6 | 33.6 | 3.1 | 26.9 | 27.0 | 2.6 | 26.6 | 31.6 | 2.1 | 17.6 | 24.0 | 5.9 |
| Motion Blur ○ | 51.7 | 49.1 | 36.7 | 50.4 | 49.9 | 37.4 | 52.5 | 56.5 | 44.1 | 56.7 | 56.0 | 48.7 |
| Crosstalk ○ | 18.4 | 16.9 | 30.0 | 21.2 | 16.5 | 27.3 | 19.3 | 13.7 | 31.9 | 24.5 | 18.7 | 41.0 |
| Incomplete Echo ○ | 59.3 | 60.7 | 65.6 | 58.0 | 61.0 | 67.8 | 60.8 | 66.7 | 70.0 | 66.9 | 66.9 | 63.3 |
| Average ● | 44.5 | 43.8 | 42.0 | 44.1 | 43.1 | 41.7 | 45.4 | 47.1 | 44.5 | 48.6 | 48.6 | 47.3 |

Table 3: **Performance evaluations of 3D object detection** under clean and adverse conditions. For each LiDAR placement strategy, we report the mAP scores (↑) for the *car* class, represented in percentage (%). The average scores only include adverse scenarios.

| Method | Center | | | Line | | | Pyramid | | | Square | | |
|---|---|---|---|---|---|---|---|---|---|---|---|---|
| | Pillar | Center | BEV | Pillar | Center | BEV | Pillar | Center | BEV | Pillar | Center | BEV |
| Clean ● | 46.5 | 55.8 | 52.5 | 43.4 | 54.0 | 49.3 | 46.1 | 55.9 | 51.0 | 43.8 | 54.0 | 49.2 |
| Fog ○ | 17.1 | 23.2 | 19.2 | 15.1 | 20.2 | 18.6 | 17.2 | 26.0 | 20.8 | 17.2 | 23.4 | 20.7 |
| Wet Ground ○ | 36.3 | 47.3 | 36.8 | 39.6 | 49.2 | 38.0 | 38.7 | 49.6 | 38.1 | 40.0 | 50.2 | 39.4 |
| Snow ○ | 37.4 | 18.9 | 27.0 | 33.6 | 22.8 | 12.2 | 36.1 | 21.1 | 15.0 | 32.5 | 19.2 | 7.3 |
| Motion Blur ○ | 27.1 | 27.3 | 12.8 | 27.1 | 9.7 | 23.6 | 29.2 | 28.6 | 29.1 | 26.1 | 13.0 | 23.6 |
| Crosstalk ○ | 25.7 | 31.6 | 8.3 | 16.9 | 12.0 | 17.2 | 26.3 | 24.9 | 23.7 | 22.3 | 14.0 | 20.1 |
| Incomplete Echo ○ | 26.2 | 22.3 | 20.9 | 25.2 | 21.5 | 21.6 | 25.8 | 25.9 | 21.6 | 25.7 | 24.6 | 22.7 |
| Average ● | 28.3 | 28.4 | 20.8 | 26.3 | 22.6 | 21.9 | 28.9 | 29.4 | 24.7 | 27.3 | 24.1 | 22.3 |

| Method | Trapezoid | | | Line-Roll | | | Pyramid-Roll | | | **Ours** | | |
|---|---|---|---|---|---|---|---|---|---|---|---|---|
| | Pillar | Center | BEV | Pillar | Center | BEV | Pillar | Center | BEV | Pillar | Center | BEV |
| Clean ● | 43.5 | 56.4 | 50.2 | 44.6 | 55.2 | 50.8 | 46.1 | 56.2 | 50.7 | 46.8 | 57.1 | 53.0 |
| Fog ○ | 16.0 | 22.1 | 19.2 | 15.2 | 19.6 | 15.2 | 17.5 | 24.8 | 22.8 | 18.3 | 22.3 | 21.8 |
| Wet Ground ○ | 40.0 | 51.7 | 39.2 | 40.3 | 49.6 | 38.3 | 39.1 | 49.2 | 40.3 | 40.1 | 51.1 | 41.3 |
| Snow ○ | 31.3 | 14.6 | 19.8 | 33.7 | 16.0 | 11.2 | 33.7 | 15.9 | 5.2 | 36.4 | 23.1 | 15.9 |
| Motion Blur ○ | 25.9 | 15.3 | 26.9 | 26.9 | 15.5 | 19.5 | 27.3 | 29.0 | 30.3 | 26.5 | 27.7 | 32.0 |
| Crosstalk ○ | 18.6 | 11.6 | 27.2 | 20.0 | 9.5 | 10.7 | 25.1 | 24.0 | 22.9 | 26.4 | 29.0 | 22.9 |
| Incomplete Echo ○ | 24.9 | 23.9 | 22.6 | 25.3 | 23.2 | 21.9 | 25.2 | 25.5 | 21.8 | 25.8 | 23.6 | 22.7 |
| Average ● | 26.1 | 23.2 | 25.8 | 26.9 | 22.2 | 19.5 | 28.0 | 28.1 | 23.9 | 28.9 | 29.5 | 26.1 |

**Detector and Segmentor.** For the benchmark, we conduct experiments using four LiDAR semantic segmentation models, *i.e.*, MinkUNet [18], SPVCNN [86], PolarNet [111], and Cylinder3D [117], and four 3D object detection models, *i.e.*, PointPillars [51], CenterPoint [107], BEVFusion-L [63], and FSTR [109]. The detailed training configurations are included in the Appendix Section B.3.

## 4.2 Comparative Study

We conduct benchmark studies to evaluate the performance of varied LiDAR placements in both clean and adverse conditions. We extensively examine the correlation between the proposed surrogate metric, known as M-SOG, and the final performance results. Through our analysis, we are able to demonstrate the effectiveness and robustness of the entire Place3D framework.

**Effectiveness of M-SOG Surrogate Metric in Place3D.** In Table 1, we report the scores of the M-SOG surrogate metric for various LiDAR placements. In Tables 2 and 3, we benchmark the 3D object detection and LiDAR semantic segmentation performance of varied LiDAR placements with state-of-the-art algorithms. Figure 4 illustrates the correlation between the proposed M-SOG

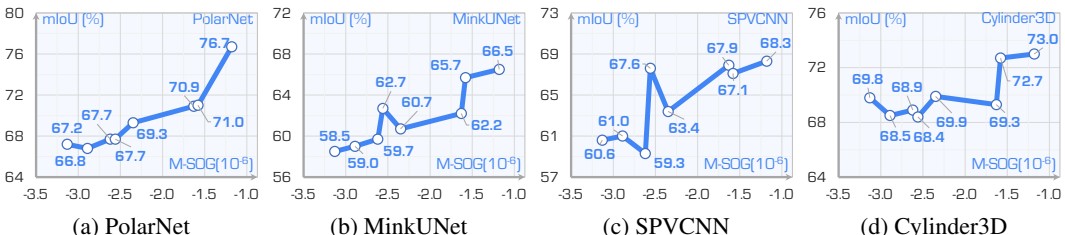

(a) PolarNet      (b) MinkUNet      (c) SPVCNN      (d) Cylinder3D

Figure 4: The correlation between M-SOG and LiDAR semantic segmentation [18, 86, 111, 117] models performance in the *clean* condition.

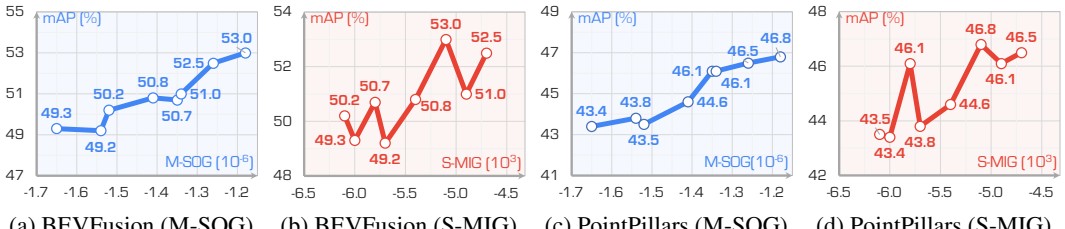

(a) BEVFusion (M-SOG)    (b) BEVFusion (S-MIG)    (c) PointPillars (M-SOG)    (d) PointPillars (S-MIG)

Figure 5: Comparisons of M-SOG with S-MIG [34] using BEVFusion-L [63] and PointPillars [51].

surrogate metric and perception capacity. The results demonstrate a clear correlation, where the performance generally improves as the M-SOG increases. While there might be fluctuations in some placements with specific algorithms, the overall relationship follows a linear correlation, highlighting the effectiveness of our M-SOG for computationally efficient sensor optimization purposes. We show more plots in the Appendix Sections C.1 and C.2.

**Comparisons to Existing Sensor Placement Methods.** The effectiveness of LiDAR sensor placement metrics lies in the linear correlation between metrics and actual performance. We set the same ROI size and conduct a quantitative comparison with *S-MIG* [34] on 3D detection (as *S-MIG* [34] only works for 3D detection task). As shown in Figure 5, our metric exhibits better linear correlation. Since we introduce semantic occupancy information as a prior for the evaluation metric, our method more accurately characterizes the boundaries in the 3D semantic scene and addresses the issue of objects and environment occlusions when using 3D bounding boxes as priors. Moreover, our LiDAR placement method makes the first attempt to include both detection and segmentation tasks. We present additional comparisons in the Appendix Section C.3.

**Superiority of Optimization via Place3D.** As shown in Tables 2 and 3, our optimized configurations achieved the best performance in both segmentation and detection tasks under both clean and adverse conditions among all models. We report per-class quantitative results in the Appendix Sections C.1 and C.2. The improvement from optimization remains significant even when comparing our optimized configurations against the best-performing baseline in each task. For segmentation, optimized placements outperform the best-performing baseline by up to 0.8% on clean datasets and by as much as 1.5% on corruption. For detection, optimized configurations exceed the best-performing baseline by up to 0.7% on clean datasets and by up to 0.3% on corruption.

**Robustness of Optimization via Place3D.** The M-SOG utilizes semantic occupancy as prior knowledge, which is invariant to changing conditions, thereby enabling robustness of optimized placement in adverse weather. Under the corrupted setting, although the correlation between M-SOG and perception performance is not as obvious as that in the clean setting and shows some fluctuation, our optimized LiDAR configuration consistently maintained its performance in adverse conditions, mirroring its effectiveness in the clean condition. While the top-performing baseline LiDAR configurations in clean datasets might be notably worse compared to others when faced with corruption, the optimized configuration via Place3D consistently shows the best performance under adverse conditions. We showcase several qualitative results on this aspect in the Appendix Section D.

**Intuitive Interpretation.** We observe several intuitive reasons why refined sensor placement is beneficial in our experiments. 1) LiDAR heights. Increasing the average height of the 4 LiDARs improves performance, likely due to an expanded field of view. 2) Variation in heights. A greater

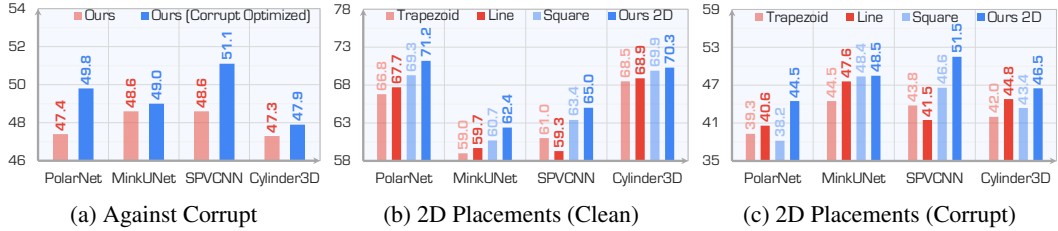

| (a) Against Corrupt | (b) 2D Placements (Clean) | (c) 2D Placements (Corrupt) |

Figure 6: Ablation results (mIoU) of placement strategies on segmentation models [111, 18, 86, 117].

difference in LiDAR heights enhances perception, as varied height distributions capture richer features from the sides of objects. 3) Uniform distribution. The pyramid placement performs better in segmentation, as a more spread-out and uniform distribution of 4 LiDARs captures a detailed surface.

## 4.3 Ablation Study

In this section, we further analyze the interplay between our proposed optimization strategy and perception performance to address two questions: 1) *How does our optimization strategy inform LiDAR placements to improve robustness against various forms of corruption?* 2) *How can our optimization strategy enhance perception performance in scenarios with limited placement options?*

**Optimizing Against Corruptions.** In the previous subsection, we deploy the placement optimized on the clean point clouds directly on corrupted ones to assess the robustness. To further showcase the capability of our method against corruptions, we compute the M-SOG scores utilizing the P-SOG derived from the corrupted semantic point clouds under adverse conditions and perform the optimization process, as shown in Figure 6a. Quantitatively compared with *Ours* in Table 2, the configurations derived from optimization on adverse data outperform the configurations generated solely on clean data. These results indicate that customizable optimization tailored for adverse settings is an effective approach to enable robust 3D perception.

**Optimizing 2D Placements.** Placing LiDARs at different heights on the roof might affect automotive aesthetics and aerodynamics. Therefore, we investigate the effect of our optimization algorithm in the presence of constraints, by limiting the LiDARs to the same height and considering only 2D placements. We fix the height of all LiDARs to find the optimal placement on the horizontal plane of the vehicle roof. We use the *Line*, *Square*, and *Trapezoid* placements in Figure 2 for comparison. Experimental results in Figures 6b and 6c show that our optimized LiDAR placements outperform baseline placements at the same LiDAR height on both clean and corruption settings.

**Influence of LiDAR Roll Angles.** Fine-tuning the orientation angles of the LiDARs on autonomous vehicles is an effective strategy to minimize blind spots and broaden the perception range. For 3D segmentation, *Pyramid-roll* slightly outperforms *Pyramid*, whereas *Line-roll* falls short of *Line*, which is highly aligned with scores of M-SOG in Table 1. For 3D detection tasks, the situation is reversed, but still consistent with the M-SOG results. This suggests that adjusting the LiDAR's angle can have varying impacts on the performance of 3D object detection and LiDAR semantic segmentation.

## 5 Conclusion

In this work, we presented Place3D, a comprehensive and systematic LiDAR placement evaluation and optimization framework. We introduced the Surrogate Metric of Semantic Occupancy Grids (M-SOG) as a novel metric to assess the impact of various LiDAR placements. Building on this metric, we culminated an optimization approach for LiDAR configuration that significantly enhances LiDAR semantic segmentation and 3D object detection performance. We validate the effectiveness of our optimization approach through a multi-condition multi-LiDAR point cloud dataset and establish a comprehensive benchmark for evaluating both baseline and our optimized LiDAR placements on detection and segmentation in diverse conditions. Our optimized placements demonstrate superior robustness and perception capabilities, outperforming all baseline configurations. By shedding light on refining the robustness of LiDAR placements for both tasks under diverse driving conditions, we anticipate that our work will pave the way for further advancements in this field of research.

## Acknowledgments

This work was supported by the Robotics Department at the University of Michigan, Ann Arbor.

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

## Appendix

## A    The Place3D Dataset

In this section, we present additional information on the statistics, features, and implementation details of the proposed Place3D dataset.

### A.1    Statistics

Our dataset consists of a total of eleven LiDAR placements, in which seven baselines are inspired by existing self-driving configurations from autonomous vehicle companies, and four LiDAR placements are obtained by optimization. Each LiDAR placement contains four LiDAR sensors. For each LiDAR configuration, the sub-dataset consists of 13,600 frames of samples, comprising 11,200 samples for training and 2,400 samples for validation, following the split ratio used in nuScenes [7]. We combined every 40 frames of samples into one scene, with a time interval of 0.5 seconds between each frame sample.

### A.2    Data Collection

In this section, we elaborate on more details of the simulator setup and LiDAR setup procedures used in collecting the Place3D dataset.

#### A.2.1    Simulator Setup

We choose four maps (Towns 1, 3, 4, and 6, *cf.* Figure 7) in CARLA v0.9.10 to collect point cloud data and generate ground truth information. For each map, we manually set six ego-vehicle routes to cover all roads with no roads overlapped. The frequency of the simulation is set to 20 Hz.

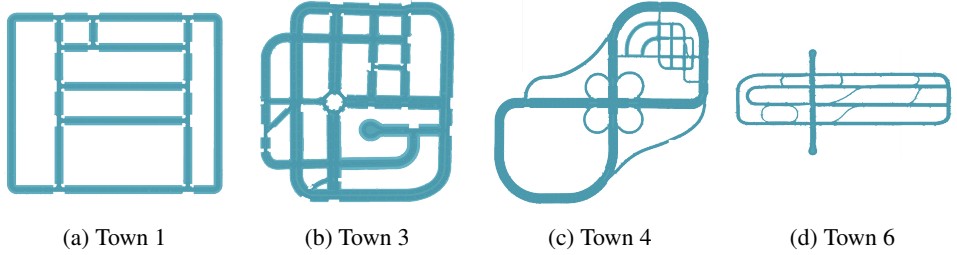

| (a) Town 1 | (b) Town 3 | (c) Town 4 | (d) Town 6 |

Figure 7: The maps used to collect the Place3D data in CARLA v0.9.10.

Table 4: Comparisons between the established benchmark and popular driving perception datasets with support for both the LiDAR semantic segmentation and 3D object detection tasks.

|  | nuScenes | Waymo | SemKITTI | Place3D |
|---|---|---|---|---|
| Detection Classes | 10 | 2 | 3 | 3 |
| Segmentation Classes | 16 | 23 | 19 | 21 |
| # of 3D Boxes | 12M | 1.4M | 80K | 71K |
| Points Per Frame | 34K | 177K | 120K | 16K |
| # of LiDAR Channels | $1 \times 32$ | $1 \times 64$ | $1 \times 64$ | $4 \times 16$ |
| Vertical FOV | $[-30.0, 10.0]$ | $[-17.6, 2.4]$ | $[-24.8, 2.0]$ | $[-24.8, 2.0]$ |
| Placement Strategy | Single | Single | Single | Multiple |
| Adverse Conditions | No | No | No | Yes |

Table 5: The attributes of the semantic LiDAR sensors used for acquiring the data.

| Attribute | Value | Attribute | Value |
|---|---|---|---|
| Channels | 16 | Upper FOV | 2.0 degrees |
| Range | 100.0 meters | Lower FOV | $-24.8$ degrees |
| Points Per Second | $5,000 \times$ channels | Horizontal FOV | 360.0 degrees |
| Rotation Frequency | 20.0 Hz | Sensor Tick | 0.5 second |

### A.2.2   LiDAR Setup

LiDAR point cloud data is collected every 0.5 simulator seconds utilizing the *Semantic LIDAR sensor* in CARLA. The attributes of *Semantic LIDAR sensor* are listed in Table 5. Notably, we utilize LiDARs with relatively low resolution to increase the challenge in object detection and semantic segmentation, leading to lower scores than those on the nuScenes [7] leaderboard. This is based on the observation that LiDAR point clouds become sparse when detecting distant objects and the use of multiple LiDARs is specifically to enhance the perception of sparse point clouds at long distances in practical applications. Our experiments with low-resolution LiDARs allow for a more evident assessment of the impact of LiDAR placements on the perception of sparse point clouds.

### A.3   Label Mappings

In this section, we provide more details for the class definitions of the LiDAR semantic segmentation and 3D object detection tasks.

### A.3.1   LiDAR Semantic Segmentation

There are a total of **21** semantic categories in our LiDAR semantic segmentation dataset, which are [1]*Building*, [2]*Fence*, [3]*Other*, [4]*Pedestrian*, [5]*Pole*, [6]*Road Line*, [7]*Road*, [8]*Sidewalk*, [9]*Vegetation*, [10]*Vehicle*, [11]*Wall*, [12]*Traffic Sign*, [13]*Ground*, [14]*Bridge*, [15]*Rail Track*, [16]*Guard Rail*, [17]*Traffic Light*, [18]*Static*, [19]*Dynamic*, [20]*Water*, and [21]*Terrain*. Table 6 presents the detailed definition of each semantic class in Place3D. Additionally, we include a *Unlabeled* tag to denote elements that have not been categorized.

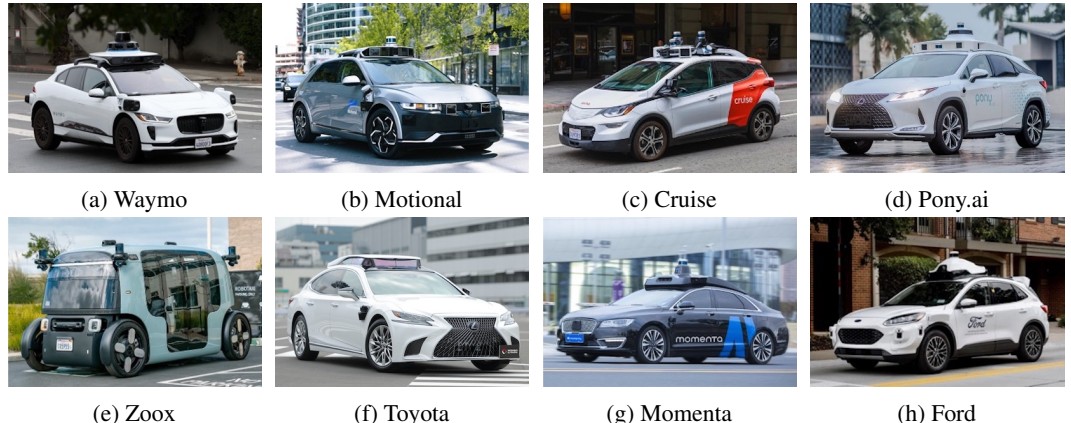

| (a) Waymo | (b) Motional | (c) Cruise | (d) Pony.ai |
| (e) Zoox | (f) Toyota | (g) Momenta | (h) Ford |

Figure 8: A diverse spectrum of existing multi-LiDAR placements employed by major autonomous vehicle companies [91, 72, 22, 76, 119, 89, 71, 27]. Images adopted from original websites.

### A.3.2    3D Object Detection

We set **3** types of common objects in the traffic scene for the 3D object detection tasks, *i.e.*, [1]*Car*, [2]*Bicyclist*, and [3]*Pedestrian*, following the same configuration as KITTI [29].

### A.4    Adverse Conditions

As shown in Figure 9 and Figure 10, the adverse conditions in the Place3D dataset can be categorized into three distinct groups:

- **Severe weather conditions**, including "fog", which causes back-scattering and attenuation of LiDAR points due to water particles in the air; "snow", where snow particles intersect with laser beams, affecting the beam's reflection and causing potential occlusions; and "wet ground", where laser pulses lose energy upon hitting wet surfaces, leading to attenuated echoes. These are commonly occurring corruptions in real driving conditions.

- **External disturbances** like "motion blur", which results from vehicle movement that causes blurring in the data, especially on bumpy surfaces or during rapid turns.

- **Internal sensor failure**, such as "crosstalk", where the light impulses from multiple sensors interfere with each other, creating noisy points; as well as "incomplete echo", where dark-colored objects result in incomplete LiDAR readings.

### A.5    License

The **Place3D** dataset is released under the *CC BY-NC-SA 4.0* license[1].

## B    Additional Implementation Detail

In this section, we present additional implementation details to facilitate the reproduction of the data generation, placement optimization, and performance evaluations in this work.

### B.1    LiDAR Placement

We adopt five commonly employed heuristic LiDAR placements, which have been adopted by several leading autonomous driving companies (see Figure 8), as our baseline. We create another two baseline placements by adjusting the rolling angles of LiDAR sensors. We obtain two placements in optimization and two placements in the ablation study. All configurations are presented in Table 7.

---

[1]https://creativecommons.org/licenses/by-nc-sa/4.0/legalcode.en.

Table 6: Summary of the semantic categories and their semantic coverage in the Place3D dataset.

| Class | ID | Description |
|---|---|---|
| Building ■ | 1 | Houses, skyscrapers, etc., and the manmade elements attached to them, such as scaffolding, awning, and ladders. |
| Fence ■ | 2 | Barriers, railing, and other upright structures that are made by wood or wire assemblies and enclose an area of ground. |
| Other ■ | 3 | Everything that does not belong to any other category. |
| Pedestrian ■ | 4 | Humans that walk, ride, or drive any kind of vehicle or mobility system, such as bicycles, scooters, skateboards, wheelchairs, etc.. |
| Pole ■ | 5 | Small mainly vertically oriented poles, such as sign poles and traffic light poles. |
| Road Line ■ | 6 | Markings on the road. |
| Road ■ | 7 | Part of ground on which vehicles usually drive. |
| Sidewalk ■ | 8 | Part of ground designated for pedestrians or cyclists, which is delimited from the road by some obstacle, such as curbs or poles. |
| Vegetation ■ | 9 | Trees, hedges, and all other types of vertical vegetation. |
| Vehicle ■ | 10 | Cars, vans, trucks, motorcycles, bikes, buses, and trains. |
| Wall ■ | 11 | Individual standing walls that are not part of a building. |
| Traffic Sign ■ | 12 | Signs installed by the city authority and are usually for traffic regulation. |
| Ground ■ | 13 | Horizontal ground-level structures that do not match any other category. |
| Bridge ■ | 14 | The structure of the bridge. |
| Rail Track ■ | 15 | All types of rail tracks that are not driveable by cars. |
| Guard Rail ■ | 16 | All types of guard rails and crash barriers. |
| Traffic Light ■ | 17 | Traffic light boxes without their poles. |
| Static ■ | 18 | Elements in the scene and props that are immovable, such fire hydrants, fixed benches, fountains, bus stops, etc. |
| Dynamic ■ | 19 | Elements whose position is susceptible to change over time, such as movable trash bins, buggies, bags, wheelchairs, animals, etc. |
| Water ■ | 20 | Horizontal water surfaces, such as lakes, sea, and rivers. |
| Terrain ■ | 21 | Grass, ground-level vegetation, soil, and sand. |

Table 7: The configuration coordinates of different LiDAR placement strategies in this work with respect to their ego-vehicle coordinate frames from the four LiDAR sensors.

| Placement | LiDAR #1 | | | | LiDAR #2 | | | | LiDAR #3 | | | | LiDAR #4 | | | |
|---|---|---|---|---|---|---|---|---|---|---|---|---|---|---|---|---|
| | $x$ | $y$ | $z$ | roll | $x$ | $y$ | $z$ | roll | $x$ | $y$ | $z$ | roll | $x$ | $y$ | $z$ | roll |
| Center | 0.0 | 0.0 | 2.2 | 0.0 | 0.0 | 0.0 | 2.4 | 0.0 | 0.0 | 0.0 | 2.6 | 0.0 | 0.0 | 0.0 | 2.8 | 0.0 |
| Line | 0.0 | -0.6 | 2.2 | 0.0 | 0.0 | -0.4 | 2.2 | 0.0 | 0.0 | 0.4 | 2.2 | 0.0 | 0.0 | 0.6 | 2.2 | 0.0 |
| Pyramid | -0.2 | -0.6 | 2.2 | 0.0 | 0.4 | 0.0 | 2.4 | 0.0 | -0.2 | 0.0 | 2.6 | 0.0 | -0.2 | 0.6 | 2.2 | 0.0 |
| Square | -0.5 | 0.5 | 2.2 | 0.0 | -0.5 | -0.5 | 2.2 | 0.0 | 0.5 | 0.5 | 2.2 | 0.0 | 0.5 | -0.5 | 2.2 | 0.0 |
| Trapezoid | -0.4 | 0.2 | 2.2 | 0.0 | -0.4 | -0.2 | 2.2 | 0.0 | 0.2 | 0.5 | 2.2 | 0.0 | 0.2 | -0.5 | 2.2 | 0.0 |
| Line-roll | 0.0 | -0.6 | 2.2 | -0.3 | 0.0 | -0.4 | 2.2 | 0.0 | 0.0 | 0.4 | 2.2 | 0.0 | 0.0 | 0.6 | 2.2 | -0.3 |
| Pyramid-roll | -0.2 | -0.6 | 2.2 | -0.3 | 0.4 | 0.0 | 2.4 | 0.0 | -0.2 | 0.0 | 2.6 | 0.0 | -0.2 | 0.6 | 2.2 | -0.3 |
| Ours-det | 0.5 | 0.5 | 2.5 | -0.3 | -0.4 | 0.1 | 2.6 | -0.2 | 0.0 | 0.0 | 2.8 | 0.0 | -0.1 | -0.5 | 2.7 | 0.0 |
| Ours-seg | 0.0 | 0.5 | 2.6 | -0.3 | 0.6 | 0.3 | 2.8 | 0.0 | 0.4 | 0.0 | 2.5 | 0.0 | 0.1 | -0.6 | 2.8 | 0.2 |
| Corruption | 0.4 | 0.5 | 2.6 | -0.3 | 0.5 | -0.4 | 2.7 | 0.0 | -0.4 | -0.3 | 2.7 | 0.1 | -0.3 | 0.5 | 2.7 | 0.0 |
| 2D-plane | 0.6 | 0.6 | 2.2 | 0.0 | 0.5 | -0.4 | 2.2 | 0.0 | -0.5 | -0.6 | 2.2 | 0.0 | -0.6 | 0.3 | 2.2 | 0.0 |

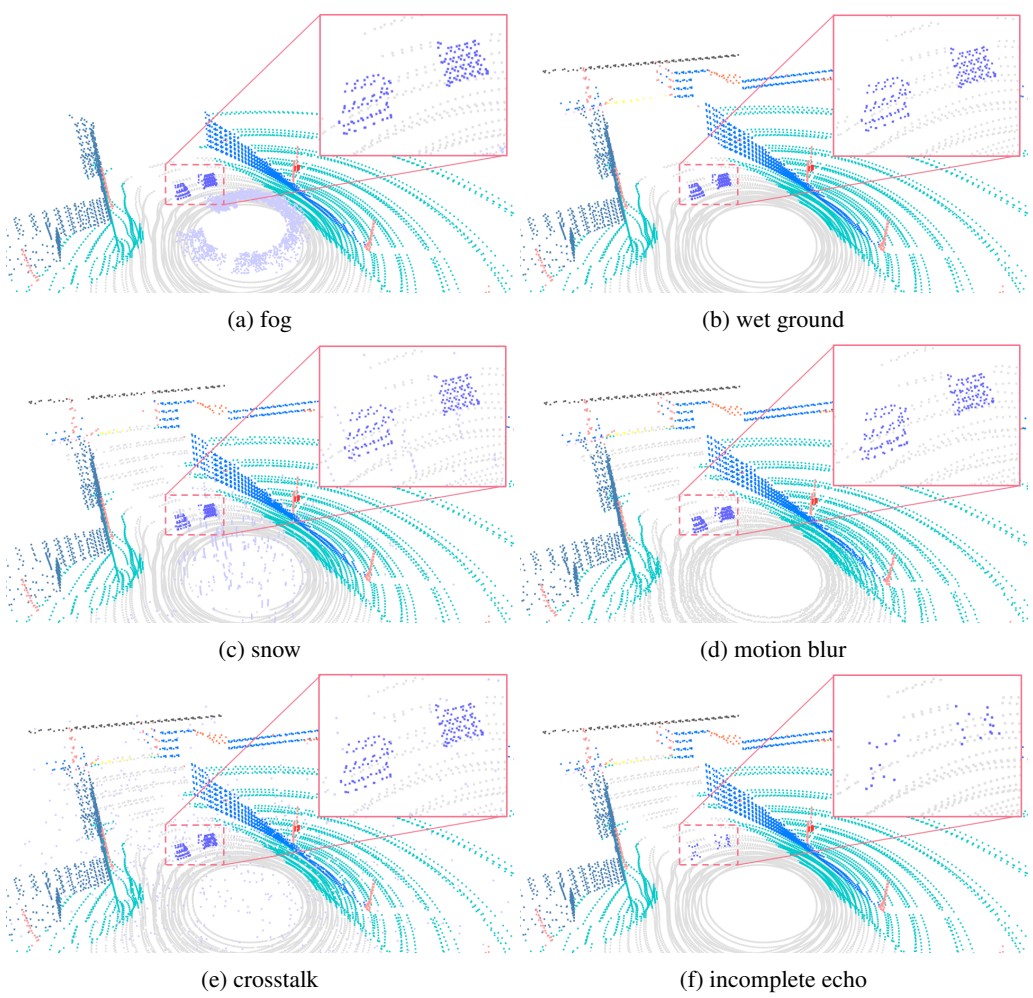

(a) fog

(b) wet ground

(c) snow

(d) motion blur

(e) crosstalk

(f) incomplete echo

Figure 9: Visual examples of LiDAR point clouds under adverse conditions in Place3D.

## B.2 Corruption Generation

In this work, we adopt the public implementations from Robo3D [46] to generate corrupted point clouds. We follow the default configuration of generating nuScenes-C to construct the adverse condition sets in Place3D. Specifically, for "fog" generation, the attenuation coefficient is randomly sampled from $[0, 0.005, 0.01, 0.02, 0.03, 0.06]$ and the back-scattering coefficient is set as $0.05$. For "wet ground" generation, the water height is set as $1.0$ millimeter. For "snow" generation, the value of snowfall rate parameter is set to $1.0$. For "motion blur" generation, the jittering noise level is set as $0.30$. For "crosstalk" generation, the disturb noise parameter is set to $0.07$. For "incomplete echo" generation, the attenuation parameter is set to $0.85$.

## B.3 Hyperparameters

In this work, we build the LiDAR placement benchmark using the MMDetection3D codebase [20]. Unless otherwise specified, we follow the conventional setups in MMDetection3D when training and evaluating the LiDAR semantic segmentation and 3D object detection models.

The detailed training configurations of the four LiDAR semantic segmentation models, *i.e.*, MinkUNet [18], SPVCNN [86], PolarNet [111], and Cylinder3D [117], are presented in Table 8. The detailed training configurations of the four 3D object detection models, *i.e.*, PointPillars [51], CenterPoint [107], BEVFusion-L [63], and FSTR [109], are presented in Table 9.

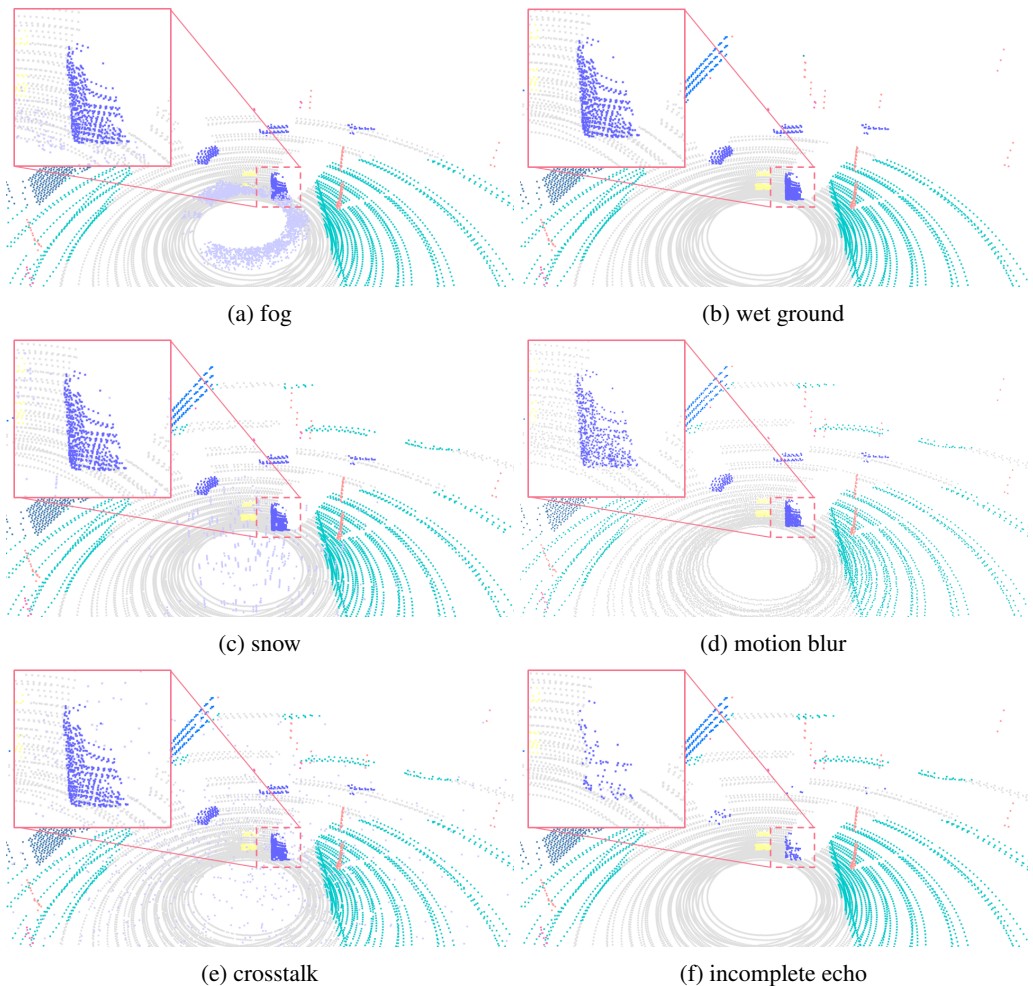

(a) fog

(b) wet ground

(c) snow

(d) motion blur

(e) crosstalk

(f) incomplete echo

Figure 10: Visual examples of LiDAR point clouds under adverse conditions in Place3D.

All LiDAR semantic segmentation models are trained and tested on eight NVIDIA A100 SXM4 80GB GPUs. All 3D object detection models are trained and tested on four NVIDIA RTX 6000 Ada 48GB GPUs. The models are evaluated on the validation sets. We do not include any type of test time augmentation or model ensembling when evaluating the models.

## C  Additional Quantitative Result

In this section, we provide additional quantitative results to better support the findings and conclusions drawn in the main body of this paper.

### C.1  LiDAR Semantic Segmentation

We provide the complete (*i.e.*, the class-wise IoU scores) results for LiDAR semantic segmentation under different LiDAR placement strategies.

We showcase the per-class LiDAR semantic segmentation results of MinkUNet [18], SPVCNN [86], PolarNet [111], and Cylinder3D [117] in Table 11. The performance of these methods is evaluated under different LiDAR placement strategies. Different LiDAR placement strategies demonstrated varying propensities towards particular classes. For instance, *Pyramid* performs well for *building*, *guard rail*, and *vegetation* compared with other placements, possibly due to an increased vertical field of view that captures these taller or layered structures more effectively. *Ours* provides the generally best balance of performance across categories, with high scores in *building*, *road*, and

Table 8: The training and optimization configurations of the four LiDAR semantic segmentation models [18, 86, 111, 117] used in our Place3D benchmark.

| Hyperparameter | MinkUNet | SPVCNN | PolarNet | Cylinder3D |
|---|---|---|---|---|
| Batch Size | $8 \times b2$ | $8 \times b2$ | $8 \times b2$ | $8 \times b2$ |
| Epochs | 50 | 50 | 50 | 50 |
| Optimizer | AdamW | AdamW | AdamW | AdamW |
| Learning Rate | $8.0e{-}3$ | $8.0e{-}3$ | $8.0e{-}3$ | $8.0e{-}3$ |
| Weight Decay | 0.01 | 0.01 | 0.01 | 0.01 |
| Epsilon | $1.0e{-}6$ | $1.0e{-}6$ | $1.0e{-}6$ | $1.0e{-}6$ |

Table 9: The training and optimization configurations of the four 3D object detection models [51, 107, 63, 109] used in our Place3D benchmark.

| Hyperparameter | PointPillars | CenterPoint | BEVFusion-L | FSTR |
|---|---|---|---|---|
| Batch Size | $4 \times b4$ | $4 \times b4$ | $4 \times b4$ | $4 \times b4$ |
| Epochs | 24 | 20 | 20 | 20 |
| Optimizer | AdamW | AdamW | AdamW | AdamW |
| Learning Rate | $1.0e{-}3$ | $1.0e{-}4$ | $1.0e{-}4$ | $1.0e{-}4$ |
| Weight Decay | 0.01 | 0.01 | 0.01 | 0.01 |
| Epsilon | $1.0e{-}6$ | $1.0e{-}6$ | $1.0e{-}6$ | $1.0e{-}6$ |

*vehicle* suggesting an effective all-around coverage for various object types. The complete benchmark results of LiDAR semantic segmentation under clean and adverse conditions are presented in Table 12. We showcase the correlation between M-SOG and LiDAR semantic segmentation performance under clean and adverse conditions in Figure 11 and 12.

We showcase the complete results of the ablation study for semantic segmentation in Table 14. we first explore the performance of our optimization algorithm under placement constraints, specifically, standardize the elevation of LiDARs to analyze their optimal positioning on the vehicle's roof's horizontal plane. Our optimized placement *2D plane* achieved better performance compared with *line*, *square*, and *trapezoid* configurations. Further, we optimize the placement for adverse conditions. While the observation suggests a trade-off in clean data compared with the *Ours*, the configuration optimized on corruption data (*corruption optimized*) achieves a significantly higher performance in adverse conditions than both baselines in Table 12 and *Ours*, indicating that custom optimization strategies, tailored for challenging conditions, represent an effective methodology to enhance robust perception capabilities in adverse environments.

## C.2 3D Object Detection

In this section, we present more experimental results for the 3D robustness evaluation of 3D object detection under adverse conditions. The performance of PointPillars [51], CenterPoint [107], BEVFusion-L [63], and FSTR [109] is evaluated under different LiDAR placement strategies in Table 13. Across all placements and conditions, there is a trend where all methods suffer a drop in performance in adverse conditions compared to clean conditions, highlighting the challenge that adverse conditions pose to 3D object detection. While some placements, like Pyramid and Square, appear to maintain relatively high performance under both clean and adverse conditions, the *Ours* placement has the highest average mAP under adverse conditions. In addition, certain placements exhibit relative strengths against specific types of corruption, which can inform the development of more resilient object detection systems for autonomous vehicles. For instance, *Pyramid* and *Pyramid-Roll*, appear to handle fog better than others, possibly due to a configuration that captures a more diverse set of angles which could mitigate the scattering effect of fog on LiDAR beams. We showcase the correlation between M-SOG and 3D Object Detection performance under clean and adverse conditions in Figure 13 and 14.

Table 10: Comparisons of the key differences among existing sensor placement approaches.

| Method | Venue | Deployment | Sensor | Prior Info | Task | Optimizing |
|---|---|---|---|---|---|---|
| S-MIG [34] | CVPR 2022 | Vehicle | LiDAR | 3D bbox | Det | ✗ |
| Jiang's [40] | ICCV 2023 | Road Side | LiDAR | 3D bbox | Det | ✓ |
| S-MS [57] | ICRA 2024 | Vehicle | LiDAR + Camera | 3D bbox | Det | ✗ |
| Place3D | Ours | Vehicle | LiDAR | Semantic Occ | Seg + Det | ✓ |

## C.3 Compare to Existing Approaches

The effectiveness of sensor placement metrics is determined by the linear correlation between these metrics and actual performance. As shown in Table 10, Jiang's method [40] is applied to roadside placements, while comparable on-vehicle methods, S-MIG [34] and S-MS [57], follow the same process for LiDARs. All these methods use 3D bounding boxes as priors to understand the 3D scene distribution. However, this approach leads to significant deviations from the actual physical boundaries of objects and fails to account for occlusion relationships between objects and the environment in LiDAR applications. Consequently, they cannot accurately describe the scene or effectively optimize LiDAR placements. Moreover, these methods are limited to evaluating LiDAR placements for detection tasks since they solely rely on 3D object distribution information.

Our method addresses these limitations by introducing semantic occupancy information as a prior for the evaluation metric. This allows for more accurate characterization of boundaries in the 3D scene and effectively addresses occlusion issues between objects and the environment. Additionally, our method leverages semantic distribution information under diverse conditions, thereby enhancing the reliability of perception in adverse weather conditions and during sensor failures.

## D Additional Qualitative Result

In this section, we provide additional qualitative examples to help visually compare different conditions presented in this work.

### D.1 Clean Condition

We showcase the qualitative results of the MinkUNet [18] model using our optimized LiDAR placement. As shown in Figure 15, the depiction across the first four rows demonstrates a commendable performance of our LiDAR placement strategy during clean conditions. This is evidenced by the predominance of gray in the error maps, which indicates a high rate of correct predictions. Also, the fifth row presents a notable exception, illustrating a failure case. This particular scene exhibits a heightened complexity with a diverse array of objects and potential occlusions that challenge predictive accuracy. The qualitative results underscore the importance of continuous refinement of LiDAR placement to achieve better perception across a comprehensive spectrum of scenarios.

### D.2 Adverse Conditions

In Figure 16, we showcase the qualitative results of MinkUNet [18] under different adverse conditions utilizing our optimized LiDAR placement. The corruptions from top to bottom rows are "fog", "wet ground", "motion blur", "crosstalk", and "incomplete echo". As can be seen, the LiDAR semantic segmentation model encounters extra difficulties when predicting under such scenarios. Erroneous predictions tend to appear in regions contaminated by different types of noises, such as airborne particles, disturbed laser reflections, and jittering scatters. It becomes apparent that enhancing the model's robustness under these adverse conditions is crucial for the practical usage of driving perception systems. To achieve better robustness, we design suitable LiDAR placements that can mitigate the degradation caused by corruptions. As discussed in Table 12, Table 13, and Table 14, our placements can largely enhance the robustness of various models.

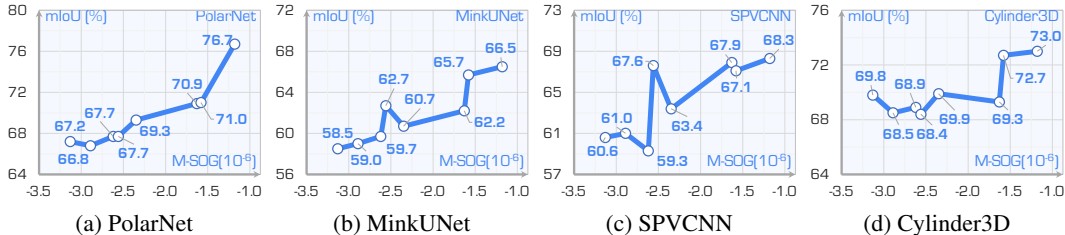

Figure 11: The correlation between M-SOG and LiDAR semantic segmentation [18, 86, 111, 117] models performance in the *clean* condition.

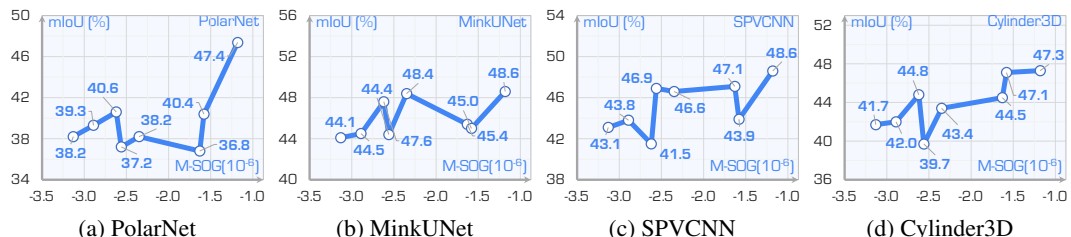

Figure 12: The correlation between M-SOG and LiDAR semantic segmentation [18, 86, 111, 117] models performance in the *adverse* condition.

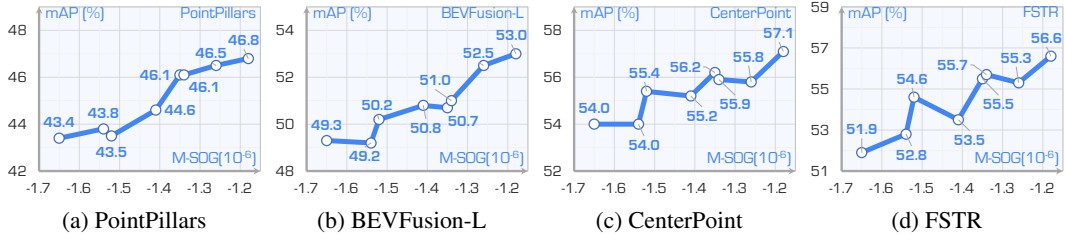

Figure 13: The correlation between M-SOG and 3D object detection [51, 107, 63, 109] models performance in the *clean* condition.

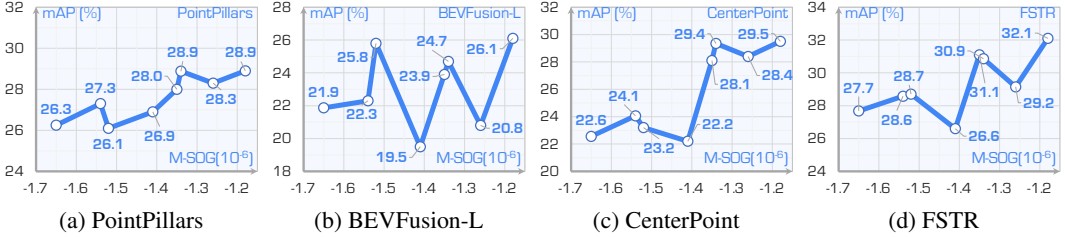

Figure 14: The correlation between M-SOG and 3D object detection [51, 107, 63, 109] models performance in the *adverse* condition.

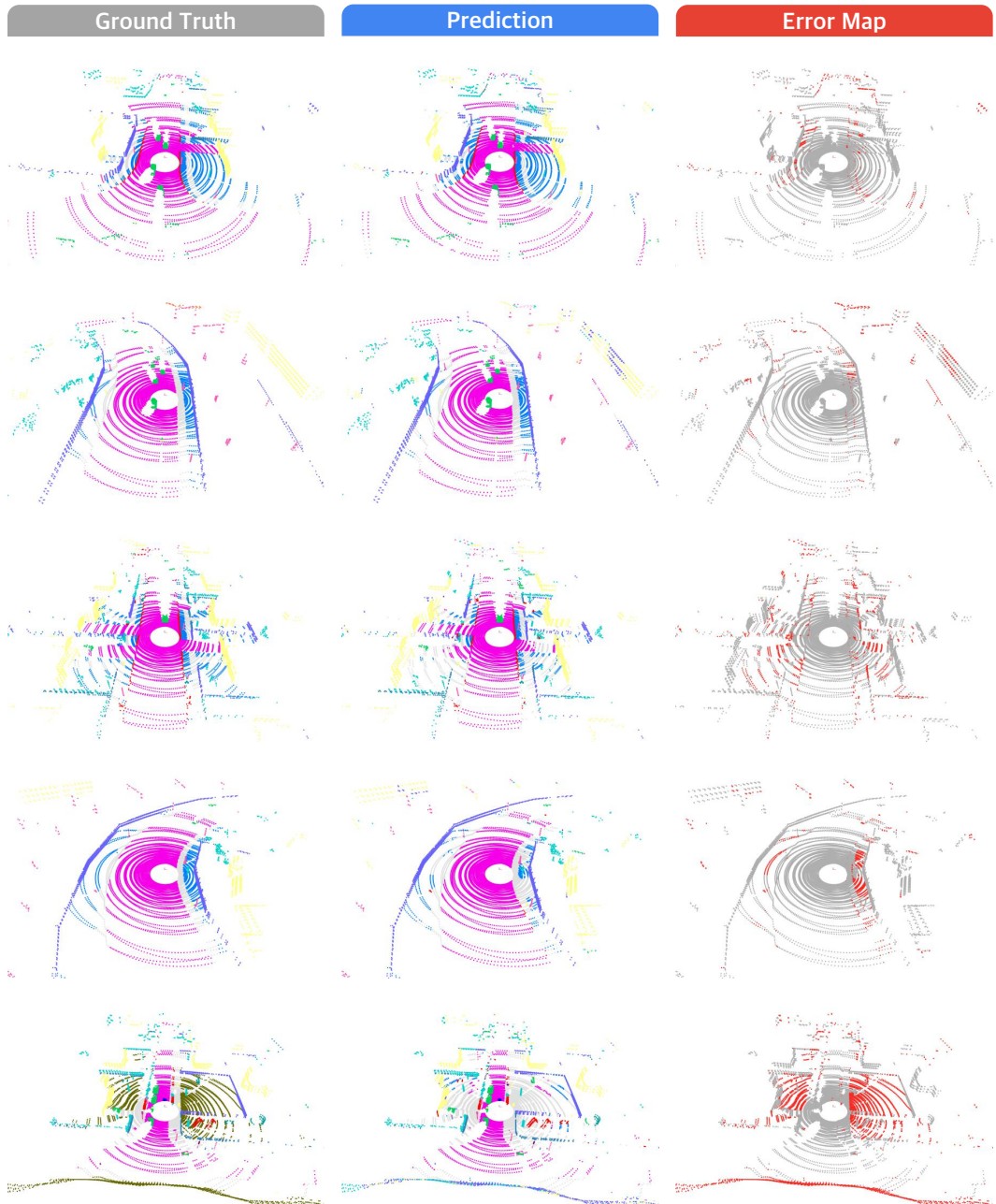

Figure 15: **Qualitative assessments** of the MinkUNet [18] model using our LiDAR placement strategy. The model is tested under the clean condition. The error maps show the correct and incorrect predictions in gray and red, respectively. Kindly refer to Table 6 for color maps. Best viewed in colors and zoomed-in for details.

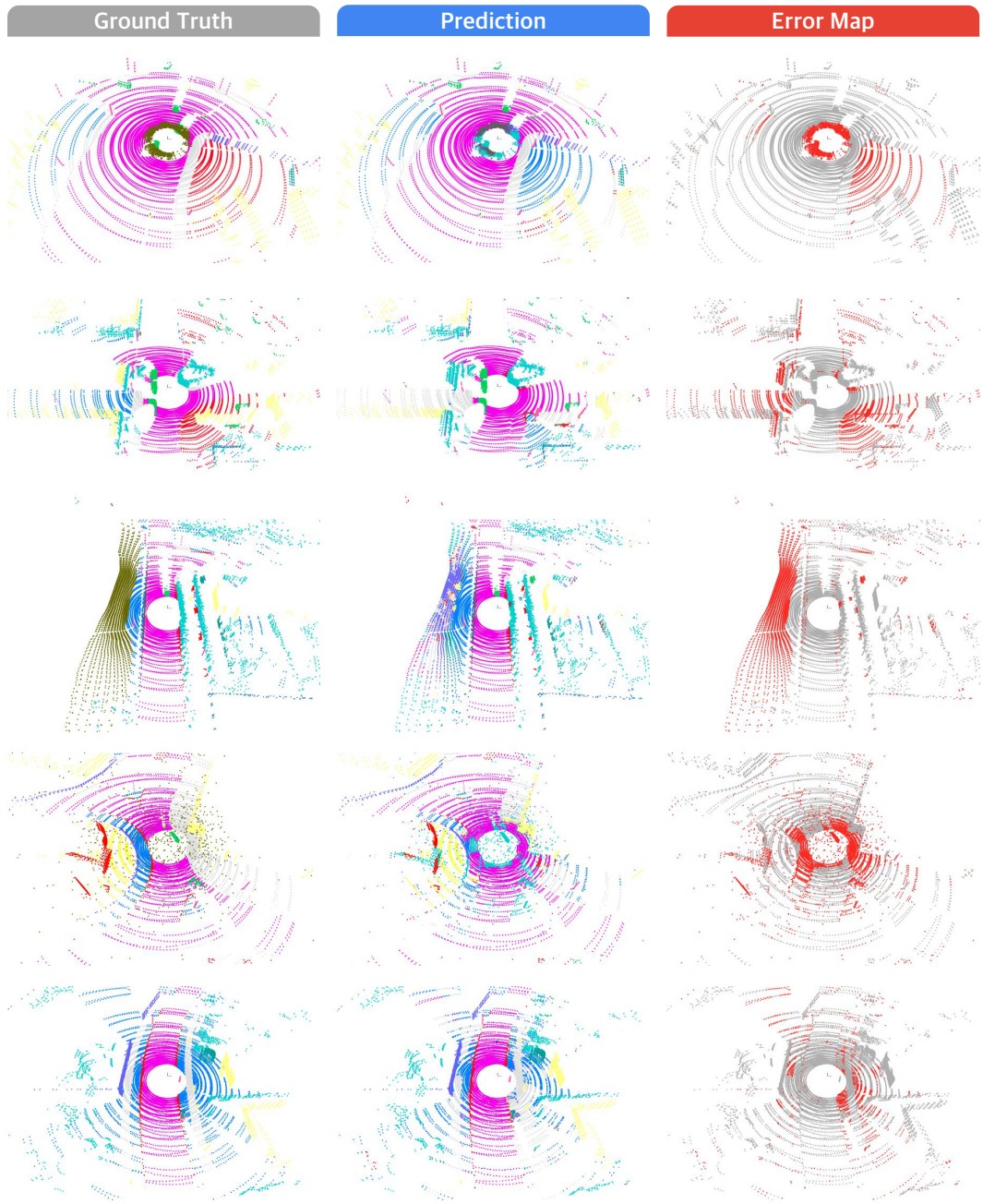

Figure 16: **Qualitative assessments** of the MinkUNet [18] model using our LiDAR placement strategy. The model is tested under adverse conditions. The error maps show the correct and incorrect predictions in gray and red, respectively. Kindly refer to Table 6 for color maps. Best viewed in colors and zoomed-in for details.

Table 11: **The per-class LiDAR semantic segmentation results** of MinkUNet [18], SPVCNN [86], PolarNet [111], and Cylinder3D [117], under different LiDAR placement strategies. All mIoU (↑) and class-wise IoU (↑) scores are given in percentage (%).

| Method | mIoU | building | fence | other | pedestrian | pole | road line | road | sidewalk | vegetation | vehicle | wall | traffic sign | ground | bridge | rail track | guard rail | traffic light | static | dynamic | water | terrain |
|---|---|---|---|---|---|---|---|---|---|---|---|---|---|---|---|---|---|---|---|---|---|---|
| **● Placement: Center** | | | | | | | | | | | | | | | | | | | | | | |
| Mink | 65.7 | 87.2 | 60.7 | 61.3 | 75.8 | 67.5 | 1.8 | 95.1 | 70.9 | 86.7 | 95.1 | 86.4 | 55.5 | 65.8 | 58.0 | 0.0 | 82.5 | 72.4 | 64.0 | 70.9 | 60.5 | 61.9 |
| SPV | 67.1 | 87.8 | 63.3 | 63.9 | 74.1 | 67.5 | 4.9 | 95.5 | 69.0 | 86.6 | 94.6 | 87.6 | 54.3 | 50.1 | 83.6 | 0.2 | 84.4 | 74.0 | 65.8 | 73.1 | 65.6 | 63.8 |
| Polar | 71.0 | 89.2 | 68.9 | 66.9 | 77.0 | 83.1 | 10.5 | 95.3 | 83.0 | 84.3 | 94.8 | 88.3 | 60.9 | 75.6 | 40.2 | 35.7 | 84.6 | 78.9 | 62.3 | 75.8 | 60.8 | 75.6 |
| Cy3D | 72.7 | 90.7 | 67.6 | 71.5 | 79.4 | 80.8 | 11.4 | 96.1 | 74.6 | 86.8 | 94.8 | 90.4 | 56.0 | 63.1 | 85.4 | 30.2 | 92.9 | 84.4 | 66.0 | 74.3 | 66.3 | 64.0 |
| Avg | 69.1 | 88.7 | 65.1 | 65.9 | 76.6 | 74.7 | 7.2 | 95.5 | 74.4 | 86.1 | 94.8 | 88.2 | 56.7 | 63.7 | 66.8 | 16.5 | 86.1 | 77.4 | 64.5 | 73.5 | 63.3 | 66.3 |
| **● Placement: Line** | | | | | | | | | | | | | | | | | | | | | | |
| Mink | 59.7 | 84.1 | 48.2 | 61.9 | 73.1 | 69.4 | 2.0 | 95.0 | 72.6 | 80.5 | 93.7 | 81.7 | 42.5 | 60.0 | 40.1 | 0.0 | 79.4 | 37.7 | 53.5 | 59.2 | 57.7 | 61.8 |
| SPV | 59.3 | 84.7 | 50.6 | 64.9 | 74.7 | 69.1 | 4.5 | 94.9 | 62.1 | 81.0 | 94.0 | 83.7 | 48.6 | 40.5 | 76.4 | 0.0 | 80.1 | 40.0 | 56.8 | 58.6 | 33.0 | 47.2 |
| Polar | 67.7 | 89.6 | 65.1 | 59.6 | 79.6 | 83.9 | 2.2 | 95.1 | 81.1 | 82.8 | 94.4 | 87.7 | 49.1 | 72.6 | 69.4 | 16.5 | 81.1 | 54.7 | 59.6 | 72.8 | 53.7 | 71.0 |
| Cy3D | 68.9 | 87.3 | 63.3 | 74.1 | 70.5 | 80.4 | 14.6 | 96.1 | 70.9 | 84.1 | 96.1 | 86.7 | 52.1 | 57.4 | 80.9 | 19.1 | 87.6 | 71.9 | 64.0 | 68.9 | 51.7 | 60.2 |
| Avg | 63.9 | 86.4 | 56.8 | 65.1 | 74.5 | 75.7 | 5.8 | 95.3 | 71.7 | 82.1 | 94.6 | 85.0 | 48.1 | 57.6 | 66.7 | 8.9 | 82.1 | 51.1 | 58.5 | 64.9 | 49.0 | 60.1 |
| **● Placement: Pyramid** | | | | | | | | | | | | | | | | | | | | | | |
| Mink | 62.7 | 87.4 | 57.6 | 64.9 | 76.5 | 73.5 | 2.4 | 94.5 | 65.1 | 82.3 | 93.7 | 85.2 | 49.6 | 47.7 | 57.1 | 0.0 | 83.5 | 65.3 | 59.1 | 60.9 | 54.2 | 55.6 |
| SPV | 67.6 | 88.2 | 60.7 | 70.2 | 78.3 | 72.6 | 9.7 | 95.5 | 74.6 | 81.7 | 92.1 | 87.4 | 55.5 | 54.5 | 86.4 | 0.0 | 81.8 | 68.1 | 64.5 | 69.2 | 62.6 | 65.9 |
| Polar | 67.7 | 90.3 | 64.7 | 55.1 | 78.6 | 84.9 | 6.2 | 95.2 | 83.3 | 84.7 | 94.8 | 87.4 | 57.8 | 74.8 | 32.2 | 28.5 | 85.8 | 45.6 | 58.9 | 78.2 | 60.9 | 74.4 |
| Cy3D | 68.4 | 87.4 | 67.8 | 70.5 | 81.2 | 82.0 | 12.7 | 95.2 | 67.4 | 84.3 | 96.4 | 86.0 | 56.4 | 54.2 | 53.5 | 20.9 | 87.1 | 81.0 | 67.3 | 71.3 | 57.3 | 55.6 |
| Avg | 66.6 | 88.3 | 62.7 | 65.2 | 78.7 | 78.3 | 7.8 | 95.1 | 72.6 | 83.3 | 94.3 | 86.5 | 54.8 | 57.8 | 57.3 | 12.4 | 84.6 | 65.0 | 62.5 | 69.9 | 58.8 | 62.9 |
| **● Placement: Square** | | | | | | | | | | | | | | | | | | | | | | |
| Mink | 60.7 | 85.3 | 51.1 | 63.7 | 75.1 | 70.6 | 3.2 | 95.5 | 64.7 | 81.6 | 95.2 | 83.7 | 46.2 | 44.8 | 63.9 | 0.0 | 79.2 | 46.3 | 54.7 | 57.6 | 59.4 | 52.0 |
| SPV | 63.4 | 85.9 | 54.3 | 69.5 | 75.8 | 70.5 | 6.0 | 95.4 | 68.9 | 82.1 | 95.5 | 84.6 | 50.1 | 53.7 | 82.8 | 0.0 | 80.2 | 40.7 | 59.2 | 63.0 | 58.4 | 54.5 |
| Polar | 69.3 | 89.9 | 64.7 | 57.2 | 79.1 | 84.0 | 3.5 | 95.3 | 82.4 | 83.9 | 95.1 | 88.9 | 50.0 | 75.6 | 75.4 | 20.2 | 86.0 | 58.2 | 58.8 | 75.8 | 58.2 | 73.8 |
| Cy3D | 69.9 | 88.7 | 63.6 | 75.1 | 80.5 | 81.5 | 13.8 | 96.2 | 76.9 | 84.9 | 97.2 | 88.0 | 55.1 | 65.9 | 51.8 | 22.3 | 90.9 | 72.5 | 67.5 | 74.9 | 55.4 | 65.2 |
| Avg | 65.8 | 87.5 | 58.4 | 66.4 | 77.6 | 76.7 | 6.6 | 95.6 | 73.2 | 83.1 | 95.8 | 86.3 | 50.4 | 60.0 | 68.5 | 10.6 | 84.1 | 54.4 | 60.1 | 67.8 | 57.9 | 61.4 |
| **● Placement: Trapezoid** | | | | | | | | | | | | | | | | | | | | | | |
| Mink | 59.0 | 84.5 | 43.8 | 64.0 | 74.4 | 70.0 | 2.4 | 95.4 | 68.7 | 81.7 | 94.6 | 82.6 | 47.1 | 54.1 | 43.7 | 0.0 | 82.6 | 52.8 | 53.9 | 54.8 | 32.1 | 56.1 |
| SPV | 61.0 | 85.8 | 54.5 | 70.0 | 73.7 | 70.4 | 7.9 | 95.1 | 64.5 | 82.2 | 94.9 | 84.4 | 49.5 | 40.2 | 74.3 | 0.0 | 76.4 | 53.6 | 60.0 | 60.4 | 30.7 | 52.1 |
| Polar | 66.8 | 90.1 | 62.6 | 59.0 | 79.3 | 83.7 | 4.5 | 95.1 | 78.0 | 83.9 | 95.2 | 87.7 | 53.4 | 74.6 | 42.0 | 18.8 | 84.7 | 54.6 | 55.7 | 74.4 | 57.2 | 67.7 |
| Cy3D | 68.5 | 88.4 | 63.1 | 75.2 | 80.3 | 80.2 | 15.1 | 96.1 | 72.6 | 83.4 | 95.8 | 87.9 | 54.5 | 61.8 | 55.7 | 13.8 | 83.9 | 77.8 | 64.6 | 66.4 | 44.8 | 58.1 |
| Avg | 63.8 | 87.2 | 56.0 | 67.1 | 76.9 | 76.1 | 7.5 | 95.4 | 71.0 | 82.8 | 95.1 | 85.7 | 51.1 | 57.7 | 53.9 | 8.2 | 81.9 | 59.7 | 58.6 | 64.0 | 46.2 | 58.5 |
| **● Placement: Line-Roll** | | | | | | | | | | | | | | | | | | | | | | |
| Mink | 58.5 | 85.3 | 47.3 | 57.9 | 73.8 | 69.6 | 2.3 | 95.1 | 71.8 | 79.2 | 91.4 | 81.6 | 47.7 | 56.7 | 22.2 | 0.0 | 78.3 | 45.5 | 54.2 | 56.1 | 53.6 | 58.2 |
| SPV | 60.6 | 85.5 | 53.6 | 62.5 | 74.3 | 70.5 | 3.4 | 95.3 | 67.1 | 81.0 | 91.6 | 84.5 | 53.5 | 49.9 | 82.6 | 0.2 | 78.3 | 44.8 | 54.2 | 61.7 | 27.4 | 51.8 |
| Polar | 67.2 | 90.6 | 65.5 | 54.1 | 79.4 | 84.5 | 2.8 | 94.8 | 81.5 | 83.0 | 94.4 | 88.2 | 50.1 | 68.9 | 39.3 | 21.2 | 87.4 | 63.8 | 56.4 | 71.6 | 61.3 | 72.9 |
| Cy3D | 69.8 | 87.1 | 58.9 | 74.5 | 80.7 | 81.8 | 13.4 | 96.5 | 76.0 | 84.1 | 96.6 | 86.0 | 55.5 | 63.7 | 62.4 | 22.8 | 90.5 | 75.9 | 66.6 | 74.9 | 55.0 | 62.0 |
| Avg | 64.0 | 87.1 | 56.3 | 62.3 | 77.1 | 76.6 | 5.5 | 95.4 | 74.1 | 81.8 | 93.5 | 85.1 | 51.7 | 59.8 | 51.6 | 11.1 | 83.6 | 57.5 | 57.9 | 66.1 | 49.3 | 61.2 |
| **● Placement: Pyramid-Roll** | | | | | | | | | | | | | | | | | | | | | | |
| Mink | 62.2 | 87.5 | 52.2 | 63.5 | 76.1 | 73.7 | 2.1 | 94.3 | 63.3 | 81.7 | 94.9 | 84.1 | 55.0 | 43.5 | 50.4 | 0.0 | 81.0 | 64.6 | 61.5 | 59.3 | 62.0 | 55.5 |
| SPV | 67.9 | 88.5 | 61.2 | 69.5 | 78.4 | 73.6 | 7.4 | 95.1 | 72.2 | 83.9 | 94.4 | 87.6 | 58.2 | 55.7 | 87.9 | 0.0 | 83.3 | 65.0 | 66.2 | 68.7 | 64.7 | 64.6 |
| Polar | 70.9 | 90.8 | 67.0 | 61.0 | 79.1 | 84.7 | 5.1 | 95.1 | 82.1 | 85.0 | 95.5 | 88.4 | 64.0 | 73.3 | 50.1 | 24.7 | 89.3 | 73.6 | 61.7 | 75.4 | 68.4 | 73.8 |
| Cy3D | 69.3 | 87.8 | 60.2 | 69.0 | 81.0 | 80.9 | 12.0 | 95.1 | 65.1 | 85.3 | 96.3 | 88.3 | 53.5 | 57.6 | 82.8 | 32.7 | 88.5 | 77.0 | 65.9 | 70.2 | 57.0 | 48.1 |
| Avg | 67.9 | 88.5 | 61.2 | 69.5 | 78.4 | 73.6 | 7.4 | 95.1 | 72.2 | 83.9 | 94.4 | 87.6 | 58.2 | 55.7 | 87.9 | 0.0 | 83.3 | 65.0 | 66.2 | 68.7 | 64.7 | 64.6 |
| **● Placement: Ours** | | | | | | | | | | | | | | | | | | | | | | |
| Mink | 66.5 | 89.6 | 56.7 | 65.1 | 73.3 | 76.4 | 4.2 | 94.6 | 70.7 | 86.0 | 94.0 | 83.8 | 62.8 | 55.9 | 49.2 | 2.6 | 85.4 | 74.6 | 63.3 | 66.5 | 78.6 | 62.4 |
| SPV | 68.3 | 89.8 | 59.5 | 66.8 | 75.2 | 75.8 | 6.0 | 94.6 | 74.2 | 85.9 | 93.4 | 85.3 | 60.1 | 58.7 | 75.5 | 0.0 | 85.5 | 76.7 | 64.9 | 65.2 | 77.3 | 64.7 |
| Polar | 76.7 | 93.5 | 70.6 | 60.7 | 79.6 | 86.1 | 8.6 | 96.0 | 86.1 | 87.2 | 94.9 | 88.8 | 68.0 | 78.8 | 56.6 | 76.1 | 89.0 | 86.1 | 63.6 | 79.1 | 82.2 | 78.2 |
| Cy3D | 73.0 | 91.7 | 66.5 | 68.3 | 80.2 | 85.1 | 12.7 | 96.1 | 82.2 | 86.9 | 95.3 | 86.7 | 54.0 | 58.4 | 28.9 | 68.2 | 90.8 | 83.9 | 73.8 | 79.3 | 74.7 | 69.3 |
| Avg | 71.1 | 91.2 | 63.3 | 65.2 | 77.1 | 80.9 | 7.9 | 95.3 | 78.3 | 86.5 | 94.4 | 86.2 | 61.2 | 63.0 | 52.6 | 36.7 | 87.7 | 80.3 | 66.4 | 72.5 | 78.2 | 68.7 |

Table 12: **Benchmark results of LiDAR semantic segmentation under clean and adverse conditions**. For each placement, we report the mIoU (↑), mAcc (↑), and ECE (↓) scores for models under the clean condition, and mIoU (↑) scores for models under adverse conditions. The mIoU and mAcc scores are given in percentage (%).

| Method | Clean | | | Adverse | | | | | | |
|---|---|---|---|---|---|---|---|---|---|---|
| | mIoU | mAcc | ECE | Fog | Wet | Snow | Blur | Cross | Echo | Avg |
| ● **Placement: Center** | | | | | | | | | | |
| MinkUNet [18] | 65.7 | 72.4 | 0.041 | 55.9 | 63.8 | 25.1 | 35.8 | 24.7 | 64.5 | 45.0 |
| SPVCNN [86] | 67.1 | 74.4 | 0.034 | 39.3 | 66.6 | 35.6 | 35.6 | 19.5 | 66.8 | 43.9 |
| PolarNet [111] | 71.0 | 76.0 | 0.033 | 43.1 | 59.2 | 11.4 | 36.1 | 23.1 | 69.4 | 40.4 |
| Cylinder3D [117] | 72.7 | 79.2 | 0.041 | 55.6 | 64.4 | 16.7 | 37.6 | 36.9 | 71.5 | 47.1 |
| ● **Placement: Line** | | | | | | | | | | |
| MinkUNet [18] | 59.7 | 67.7 | 0.037 | 51.7 | 60.2 | 35.5 | 52.0 | 27.1 | 59.2 | 47.6 |
| SPVCNN [86] | 59.3 | 66.7 | 0.068 | 42.8 | 57.9 | 31.3 | 46.1 | 13.6 | 57.1 | 41.5 |
| PolarNet [111] | 67.7 | 74.1 | 0.034 | 43.1 | 61.6 | 4.4 | 48.4 | 18.9 | 67.3 | 40.6 |
| Cylinder3D [117] | 68.9 | 76.3 | 0.045 | 55.5 | 66.4 | 4.7 | 39.4 | 34.3 | 68.3 | 44.8 |
| ● **Placement: Pyramid** | | | | | | | | | | |
| MinkUNet [18] | 62.7 | 70.6 | 0.072 | 52.9 | 60.3 | 25.2 | 50.7 | 17.3 | 60.2 | 44.4 |
| SPVCNN [86] | 67.6 | 74.0 | 0.037 | 48.6 | 66.6 | 30.2 | 55.1 | 14.8 | 65.9 | 46.9 |
| PolarNet [111] | 67.7 | 73.0 | 0.032 | 34.3 | 61.0 | 2.3 | 49.1 | 9.6 | 66.9 | 37.2 |
| Cylinder3D [117] | 68.4 | 76.0 | 0.093 | 51.0 | 52.2 | 5.0 | 42.5 | 26.6 | 60.9 | 39.7 |
| ● **Placement: Square** | | | | | | | | | | |
| MinkUNet [18] | 60.7 | 68.4 | 0.043 | 55.6 | 61.9 | 33.5 | 51.5 | 26.5 | 61.2 | 48.4 |
| SPVCNN [86] | 63.4 | 70.2 | 0.031 | 40.7 | 64.3 | 38.3 | 53.9 | 18.6 | 63.7 | 46.6 |
| PolarNet [111] | 69.3 | 74.7 | 0.033 | 39.9 | 50.0 | 6.1 | 49.1 | 15.1 | 68.8 | 38.2 |
| Cylinder3D [117] | 69.9 | 76.7 | 0.044 | 52.0 | 55.6 | 2.7 | 44.2 | 37.1 | 68.7 | 43.4 |
| ● **Placement: Trapezoid** | | | | | | | | | | |
| MinkUNet [18] | 59.0 | 66.2 | 0.040 | 49.7 | 60.4 | 27.6 | 51.7 | 18.4 | 59.3 | 44.5 |
| SPVCNN [86] | 61.0 | 68.8 | 0.044 | 40.9 | 61.3 | 33.6 | 49.1 | 16.9 | 60.7 | 43.8 |
| PolarNet [111] | 66.8 | 72.3 | 0.034 | 37.5 | 65.3 | 2.8 | 46.7 | 15.4 | 67.8 | 39.3 |
| Cylinder3D [117] | 68.5 | 75.4 | 0.057 | 52.1 | 64.6 | 3.1 | 36.7 | 30.0 | 65.6 | 42.0 |
| ● **Placement: Line-Roll** | | | | | | | | | | |
| MinkUNet [18] | 58.5 | 66.4 | 0.047 | 48.6 | 59.2 | 26.9 | 50.4 | 21.2 | 58.0 | 44.1 |
| SPVCNN [86] | 60.6 | 68.0 | 0.034 | 42.2 | 62.0 | 27.0 | 49.9 | 16.5 | 61.0 | 43.1 |
| PolarNet [111] | 67.2 | 72.8 | 0.037 | 38.2 | 62.9 | 2.2 | 46.3 | 14.2 | 65.4 | 38.2 |
| Cylinder3D [117] | 69.8 | 77.0 | 0.048 | 49.7 | 65.4 | 2.6 | 37.4 | 27.3 | 67.8 | 41.7 |
| ● **Placement: Pyramid-Roll** | | | | | | | | | | |
| MinkUNet [18] | 62.2 | 69.6 | 0.051 | 52.2 | 60.9 | 26.6 | 52.5 | 19.3 | 60.8 | 45.4 |
| SPVCNN [86] | 67.9 | 74.2 | 0.033 | 47.2 | 67.1 | 31.6 | 56.5 | 13.7 | 66.7 | 47.1 |
| PolarNet [111] | 70.9 | 75.9 | 0.035 | 36.3 | 49.1 | 2.3 | 51.4 | 13.3 | 68.6 | 36.8 |
| Cylinder3D [117] | 69.3 | 77.0 | 0.048 | 50.7 | 67.9 | 2.1 | 44.1 | 31.9 | 70.0 | 44.5 |
| ● **Placement: Ours** | | | | | | | | | | |
| MinkUNet [18] | 66.5 | 73.2 | 0.031 | 59.5 | 66.6 | 17.6 | 56.7 | 24.5 | 66.9 | 48.6 |
| SPVCNN [86] | 68.3 | 74.6 | 0.034 | 59.1 | 66.7 | 24.0 | 56.0 | 18.7 | 66.9 | 48.6 |
| PolarNet [111] | 76.7 | 81.5 | 0.033 | 57.3 | 65.8 | 2.8 | 55.0 | 27.3 | 76.1 | 47.4 |
| Cylinder3D [117] | 73.0 | 78.9 | 0.037 | 57.6 | 67.2 | 5.9 | 48.7 | 41.0 | 63.3 | 47.3 |

Table 13: **Benchmark results of 3D object detection under clean and adverse conditions**. For each placement, we report the mAP (↑) scores of three classes (*car*, *pedestrian*, and *bicyclist*) for models under the clean condition, and mAP (↑) scores of *car* for models under adverse conditions. The mAP scores are given in percentage (%).

| Method | Clean | | | Adverse | | | | | | |
|---|---|---|---|---|---|---|---|---|---|---|
| | Car | Ped | Bicy | Fog | Wet | Snow | Blur | Cross | Echo | Avg |
| • **Placement: Center** | | | | | | | | | | |
| PointPillars [51] | 46.5 | 19.4 | 27.1 | 17.1 | 36.3 | 37.4 | 27.1 | 25.7 | 26.2 | 28.3 |
| CenterPoint [107] | 55.8 | 28.7 | 28.8 | 23.2 | 47.3 | 18.9 | 27.3 | 31.6 | 22.3 | 28.4 |
| BEVFusion-L [63] | 52.5 | 31.9 | 32.2 | 19.2 | 36.8 | 27.0 | 12.8 | 8.3 | 20.9 | 20.8 |
| FSTR [109] | 55.3 | 27.7 | 29.3 | 23.8 | 44.1 | 30.7 | 25.4 | 25.0 | 25.9 | 29.2 |
| • **Placement: Line** | | | | | | | | | | |
| PointPillars [51] | 43.4 | 22.0 | 27.7 | 15.1 | 39.6 | 33.6 | 27.1 | 16.9 | 25.2 | 26.3 |
| CenterPoint [107] | 54.0 | 34.2 | 37.7 | 20.2 | 49.2 | 22.8 | 9.7 | 12.0 | 21.5 | 22.6 |
| BEVFusion-L [63] | 49.3 | 29.0 | 29.5 | 18.6 | 38.0 | 12.2 | 23.6 | 17.2 | 21.6 | 21.9 |
| FSTR [109] | 51.9 | 30.2 | 33.0 | 22.7 | 45.4 | 27.0 | 23.9 | 20.1 | 27.0 | 27.7 |
| • **Placement: Pyramid** | | | | | | | | | | |
| PointPillars [51] | 46.1 | 24.4 | 29.0 | 17.2 | 38.7 | 36.1 | 29.2 | 26.3 | 25.8 | 28.9 |
| CenterPoint [107] | 55.9 | 37.4 | 35.6 | 26.0 | 49.6 | 21.1 | 28.6 | 24.9 | 25.9 | 29.4 |
| BEVFusion-L [63] | 51.0 | 21.7 | 27.9 | 20.8 | 38.1 | 15.0 | 29.1 | 23.7 | 21.6 | 24.7 |
| FSTR [109] | 55.7 | 29.4 | 33.8 | 25.8 | 44.8 | 24.0 | 33.4 | 28.5 | 28.7 | 30.9 |
| • **Placement: Square** | | | | | | | | | | |
| PointPillars [51] | 43.8 | 20.8 | 27.1 | 17.2 | 40.0 | 32.5 | 26.1 | 22.3 | 25.7 | 27.3 |
| CenterPoint [107] | 54.0 | 35.5 | 34.1 | 23.4 | 50.2 | 19.2 | 13.0 | 14.0 | 24.6 | 24.1 |
| BEVFusion-L [63] | 49.2 | 27.0 | 26.7 | 20.7 | 39.4 | 7.3 | 23.6 | 20.1 | 22.7 | 22.3 |
| FSTR [109] | 52.8 | 30.3 | 31.3 | 23.7 | 47.2 | 23.4 | 25.1 | 23.1 | 29.0 | 28.6 |
| • **Placement: Trapezoid** | | | | | | | | | | |
| PointPillars [51] | 43.5 | 21.5 | 27.3 | 16.0 | 40.0 | 31.3 | 25.9 | 18.6 | 24.9 | 26.1 |
| CenterPoint [107] | 55.4 | 35.6 | 37.5 | 22.1 | 51.7 | 14.6 | 15.3 | 11.6 | 23.9 | 23.2 |
| BEVFusion-L [63] | 50.2 | 30.0 | 31.7 | 19.2 | 39.2 | 19.8 | 26.9 | 27.2 | 22.6 | 25.8 |
| FSTR [109] | 54.6 | 30.0 | 33.3 | 22.9 | 46.9 | 26.0 | 26.5 | 23.4 | 26.5 | 28.7 |
| • **Placement: Line-Roll** | | | | | | | | | | |
| PointPillars [51] | 44.6 | 21.3 | 27.0 | 15.2 | 40.3 | 33.7 | 26.9 | 20.0 | 25.3 | 26.9 |
| CenterPoint [107] | 55.2 | 32.7 | 37.2 | 19.6 | 49.6 | 16.0 | 15.5 | 9.5 | 23.2 | 22.2 |
| BEVFusion-L [63] | 50.8 | 29.4 | 29.5 | 15.2 | 38.3 | 11.2 | 19.5 | 10.7 | 21.9 | 19.5 |
| FSTR [109] | 53.5 | 29.8 | 32.4 | 21.0 | 46.2 | 24.2 | 24.1 | 16.2 | 27.8 | 26.6 |
| • **Placement: Pyramid-Roll** | | | | | | | | | | |
| PointPillars [51] | 46.1 | 23.6 | 27.9 | 17.5 | 39.1 | 33.7 | 27.3 | 25.1 | 25.2 | 28.0 |
| CenterPoint [107] | 56.2 | 36.5 | 35.9 | 24.8 | 49.2 | 15.9 | 29.0 | 24.0 | 25.5 | 28.1 |
| BEVFusion-L [63] | 50.7 | 22.7 | 28.2 | 22.8 | 40.3 | 5.2 | 30.3 | 22.9 | 21.8 | 23.9 |
| FSTR [109] | 55.5 | 29.9 | 32.0 | 26.3 | 47.2 | 22.9 | 33.5 | 28.1 | 28.8 | 31.1 |
| • **Placement: Ours** | | | | | | | | | | |
| PointPillars [51] | 46.8 | 24.9 | 27.2 | 18.3 | 40.1 | 36.4 | 26.5 | 26.4 | 25.8 | 28.9 |
| CenterPoint [107] | 57.1 | 34.4 | 37.3 | 22.3 | 51.1 | 23.1 | 27.7 | 29.0 | 23.6 | 29.5 |
| BEVFusion-L [63] | 53.0 | 28.7 | 29.5 | 21.8 | 41.3 | 15.9 | 32.0 | 22.9 | 22.7 | 26.1 |
| FSTR [109] | 56.6 | 31.9 | 34.1 | 25.1 | 47.5 | 28.8 | 32.1 | 30.6 | 28.2 | 32.1 |

Table 14: **Ablation study of LiDAR semantic segmentation under clean and adverse conditions**. For each placement, we report the mIoU (↑), mAcc (↑), and ECE (↓) scores for models under the clean condition, and mIoU (↑) scores for models under adverse conditions. The mIoU and mAcc scores are given in percentage (%).

| Method | Clean | | | Adverse | | | | | | |
|---|---|---|---|---|---|---|---|---|---|---|
| | mIoU | mAcc | ECE | Fog | Wet | Snow | Blur | Cross | Echo | **Avg** |
| • **Placement: Line** | | | | | | | | | | |
| MinkUNet [18] | 59.7 | 67.7 | 0.037 | 51.7 | 60.2 | 35.5 | 52.0 | 27.1 | 59.2 | 47.6 |
| SPVCNN [86] | 59.3 | 66.7 | 0.068 | 42.8 | 57.9 | 31.3 | 46.1 | 13.6 | 57.1 | 41.5 |
| PolarNet [111] | 67.7 | 74.1 | 0.034 | 43.1 | 61.6 | 4.4 | 48.4 | 18.9 | 67.3 | 40.6 |
| Cylinder3D [117] | 68.9 | 76.3 | 0.045 | 55.5 | 66.4 | 4.7 | 39.4 | 34.3 | 68.3 | 44.8 |
| • **Placement: Square** | | | | | | | | | | |
| MinkUNet [18] | 60.7 | 68.4 | 0.043 | 55.6 | 61.9 | 33.5 | 51.5 | 26.5 | 61.2 | 48.4 |
| SPVCNN [86] | 63.4 | 70.2 | 0.031 | 40.7 | 64.3 | 38.3 | 53.9 | 18.6 | 63.7 | 46.6 |
| PolarNet [111] | 69.3 | 74.7 | 0.033 | 39.9 | 50.0 | 6.1 | 49.1 | 15.1 | 68.8 | 38.2 |
| Cylinder3D [117] | 69.9 | 76.7 | 0.044 | 52.0 | 55.6 | 2.7 | 44.2 | 37.1 | 68.7 | 43.4 |
| • **Placement: Trapezoid** | | | | | | | | | | |
| MinkUNet [18] | 59.0 | 66.2 | 0.040 | 49.7 | 60.4 | 27.6 | 51.7 | 18.4 | 59.3 | 44.5 |
| SPVCNN [86] | 61.0 | 68.8 | 0.044 | 40.9 | 61.3 | 33.6 | 49.1 | 16.9 | 60.7 | 43.8 |
| PolarNet [111] | 66.8 | 72.3 | 0.034 | 37.5 | 65.3 | 2.8 | 46.7 | 15.4 | 67.8 | 39.3 |
| Cylinder3D [117] | 68.5 | 75.4 | 0.057 | 52.1 | 64.6 | 3.1 | 36.7 | 30.0 | 65.6 | 42.0 |
| • **Placement: Line-Roll** | | | | | | | | | | |
| MinkUNet [18] | 58.5 | 66.4 | 0.047 | 48.6 | 59.2 | 26.9 | 50.4 | 21.2 | 58.0 | 44.1 |
| SPVCNN [86] | 60.6 | 68.0 | 0.034 | 42.2 | 62.0 | 27.0 | 49.9 | 16.5 | 61.0 | 43.1 |
| PolarNet [111] | 67.2 | 72.8 | 0.037 | 38.2 | 62.9 | 2.2 | 46.3 | 14.2 | 65.4 | 38.2 |
| Cylinder3D [117] | 69.8 | 77.0 | 0.048 | 49.7 | 65.4 | 2.6 | 37.4 | 27.3 | 67.8 | 41.7 |
| • **Placement: Ours 2D Plane** | | | | | | | | | | |
| MinkUNet [18] | 62.4 | 71.0 | 0.059 | 56.1 | 58.7 | 32.8 | 51.9 | 25.6 | 66.1 | 48.5 |
| SPVCNN [86] | 65.0 | 71.8 | 0.050 | 56.9 | 66.1 | 37.9 | 56.4 | 26.0 | 65.7 | 51.5 |
| PolarNet [111] | 71.2 | 76.4 | 0.046 | 52.8 | 64.4 | 5.0 | 51.7 | 23.0 | 69.8 | 44.5 |
| Cylinder3D [117] | 70.3 | 75.7 | 0.047 | 52.6 | 67.7 | 5.0 | 45.9 | 36.8 | 71.1 | 46.5 |
| • **Placement: Corruption Optimized** | | | | | | | | | | |
| MinkUNet [18] | 62.9 | 70.6 | 0.062 | 57.0 | 59.9 | 31.9 | 52.4 | 25.0 | 67.6 | 49.0 |
| SPVCNN [86] | 64.8 | 72.0 | 0.051 | 56.4 | 65.6 | 36.8 | 57.0 | 25.8 | 64.8 | 51.1 |
| PolarNet [111] | 73.9 | 79.3 | 0.037 | 60.7 | 66.4 | 5.6 | 54.4 | 34.8 | 76.8 | 49.8 |
| Cylinder3D [117] | 72.5 | 79.1 | 0.040 | 54.0 | 69.9 | 5.9 | 47.3 | 37.8 | 72.7 | 47.9 |
| • **Placement: Ours** | | | | | | | | | | |
| MinkUNet [18] | 66.5 | 73.2 | 0.031 | 59.5 | 66.6 | 17.6 | 56.7 | 24.5 | 66.9 | 48.6 |
| SPVCNN [86] | 68.3 | 74.6 | 0.034 | 59.1 | 66.7 | 24.0 | 56.0 | 18.7 | 66.9 | 48.6 |
| PolarNet [111] | 76.7 | 81.5 | 0.033 | 57.3 | 65.8 | 2.8 | 55.0 | 27.3 | 76.1 | 47.4 |
| Cylinder3D [117] | 73.0 | 78.9 | 0.037 | 57.6 | 67.2 | 5.9 | 48.7 | 41.0 | 63.3 | 47.3 |

# E  Proofs

In this section, we present the full proof of Theorem 1 to certify the global optimally regarding the solved local optima. Additionally, we present the proof of Corollary 1 as a special case with bounded hyper-rectangle search space only given the Lipschitz constant of the objective function.

## E.1  Full Proof of Theorem 1

**Theorem 2** (Optimality Certification). *Given the continuous objective function $G : \mathbb{R}^n \to \mathbb{R}$ with Lipschitz constant $k_G$ w.r.t. input $\boldsymbol{u} \in \mathcal{U} \subset \mathbb{R}^n$ under $\ell_2$ norm, suppose over a $\delta$-density Grids subset $S \subset \mathcal{U}$, the distance between the maximal and minimal of function $G$ over $S$ is upper-bounded by $C_M$, and the local optima is $\boldsymbol{u}_S^* = \arg\min_{\boldsymbol{u} \in S} G(x)$, the following optimality certification regarding $x \in \mathcal{U}$ holds that:*

$$\|G(\boldsymbol{u}^*) - G(\boldsymbol{u}_S^*)\|_2 \ \leq \ C_M + k_G \delta \,, \tag{13}$$

*where $\boldsymbol{u}^*$ is the global optima over $\mathcal{U}$.*

*Proof.* Based on the sampling over subset $S \subset \mathcal{U}$ with $\ell_2$-norm density $\delta$, we have:

$$\forall \mathbf{u} \in \mathcal{U}, \min_{\mathbf{u}_S \in S} \|\mathbf{u} - \mathbf{u}_S\|_2 \ \leq \ \delta \,. \tag{14}$$

From the continuous objective function $G : \mathbb{R}^n \to \mathbb{R}$ with Lipschitz constant $k_G$ w.r.t. input $\mathbf{u} \in \mathcal{U} \subset \mathbb{R}^n$ under $\ell_2$ norm, it holds that:

$$\|G(\mathbf{u}_1) - G(\mathbf{u}_2)\|_2 \ \leq \ k_G \|\mathbf{u}_1 - \mathbf{u}_2\|_2, \ \forall \mathbf{u}_1, \mathbf{u}_2 \in \mathcal{U} \,. \tag{15}$$

Since the distance between the maximal and minimal of function $G$ over $S$ is upper-bounded by $C_M$, we then have:

$$\|G(\mathbf{u}_1) - G(\mathbf{u}_2)\|_2 \ \leq \ \|\max_{\mathbf{u} \in S} G(\mathbf{u}) - \min_{\mathbf{u} \in S} G(\mathbf{u})\|_2 \ \leq \ C_M, \forall \mathbf{u}_1, \mathbf{u}_2 \in S \,. \tag{16}$$

Then by absolute value inequality with $\mathbf{u}_S \in S \subset \mathcal{U}$ s.t. $\|\mathbf{u}^* - \mathbf{u}_S\|_2 \leq \delta$ and combining all inequalities above, we have:

$$\|G(\mathbf{u}_S^*) - G(\mathbf{u}_S^*)\|_2 \leq \|G(\mathbf{u}^*) - G(\mathbf{u}_S)\|_2 + \|G(\mathbf{u}^*) - G(\mathbf{u}_S)\|_2 \tag{17}$$
$$\leq k_G \|\mathbf{u}^* - \mathbf{u}_S\|_2 + C_M \tag{18}$$
$$\leq k_G \delta + C_M \,, \tag{19}$$

which concludes the proof. □

## E.2  Proof of Corollary 1

**Corollary 2.** *When $\mathcal{U}$ is a hyper-rectangle with the bounded $\ell_2$ norm of domain $U_i \in \mathbb{R}$ at each dimension $i$, with $i = 1, 2, \ldots, n$, Thm.2 can hold in a more general way by only assuming that the Lipschitz constant $k_G$ of the objective function is given, where the following optimality certification regarding $x \in \mathcal{U}$ holds that:*

$$\|G(\boldsymbol{u}^*) - G(\boldsymbol{u}_S^*)\|_2 \ \leq \ k_G \sum_{i=1}^{n} U_i + k_G \delta \,. \tag{20}$$

*Proof.* Similar to the proof of Thm. 2, by the sampling over subset $S \subset \mathcal{U}$ with $\ell_2$-norm density $\delta$, we have:

$$\forall \mathbf{u} \in \mathcal{U}, \ \min_{\mathbf{u}_S \in S} \|\mathbf{u} - \mathbf{u}_S\|_2 \ \leq \ \delta. \tag{21}$$

From the continuous objective function $G : \mathbb{R}^n \to \mathbb{R}$ with Lipschitz constant $k_G$ w.r.t. input $\mathbf{u} \in \mathcal{U} \subset \mathbb{R}^n$ under $\ell_2$ norm, it holds that:

$$\|G(\mathbf{u}_1) - G(\mathbf{u}_2)\|_2 \ \leq \ k_G \|\mathbf{u}_1 - \mathbf{u}_2\|_2, \ \forall \mathbf{u}_1, \mathbf{u}_2 \in \mathcal{U} \,. \tag{22}$$

Now since the input space $\mathcal{U}$ is a hyper-rectangle with the bounded $\ell_2$ norm of domain $U_i \in \mathbb{R}$ at each dimension $i$, with $i = 1, 2, \ldots, n$, for any $i$-th dimension $\mathbf{u}_1^{(i)}, \mathbf{u}_2^{(i)} \in \mathcal{U}^{(i)} \subset \mathbb{R}$, it holds that:

$$\|\mathbf{u}_1^{(i)} - \mathbf{u}_2^{(i)}\|_2 \;\leq\; U_i, \; \forall \mathbf{u}_1^{(i)}, \mathbf{u}_2^{(i)} \in \mathcal{U}^{(i)}, \; i = 1, 2, \ldots, n \;, \tag{23}$$

By summing up all dimensions with absolute value inequality at $\mathbf{u}_{max}, \mathbf{u}_{min}$, we have:

$$\| \max_{\mathbf{u} \in S} G(\mathbf{u}) - \min_{\mathbf{u} \in S} G(\mathbf{u}) \|_2 \;:=\; \|G(\mathbf{u}_{max}) - G(\mathbf{u})_{min}\|_2 \tag{24}$$

$$\leq \|G(\sum_{i=1}^{n} \mathbf{u}_{max}^{(i)}) - G(\sum_{i=1}^{n} \mathbf{u}_{min}^{(i)})\|_2 \tag{25}$$

$$\leq k_G \| \sum_{i=1}^{n} \mathbf{u}_{max}^{(i)} - \sum_{i=1}^{n} \mathbf{u}_{min}^{(i)} \|_2 \tag{26}$$

$$\leq k_G \sum_{i=1}^{n} \| \mathbf{u}_{max}^{(i)} - \mathbf{u}_{min}^{(i)} \|_2 \tag{27}$$

$$\leq k_G \sum_{i=1}^{n} U_i \;. \tag{28}$$

Since the distance between the maximal and minimal of function $G$ over $S$ is upper-bounded by $C_M$, we then have:

$$\|G(\mathbf{u}_1) - G(\mathbf{u}_2)\|_2 \;\leq\; \| \max_{\mathbf{u} \in S} G(\mathbf{u}) - \min_{\mathbf{u} \in S} G(\mathbf{u}) \|_2 \;\leq\; k_G \sum_{i=1}^{n} U_i, \; \forall \mathbf{u}_1, \mathbf{u}_2 \in S \;. \tag{29}$$

Then by absolute value inequality with $\mathbf{u}_S \in S \subset \mathcal{U}$ s.t. $\|\mathbf{u}^* - \mathbf{u}_S\|_2 \leq \delta$ and combining all inequalities above, we have:

$$\|G(\mathbf{u}^*) - G(\mathbf{u}_S^*)\|_2 \;\leq\; \|G(\mathbf{u}^*) - G(\mathbf{u}_S)\|_2 + \|G(\mathbf{u}^*) - G(\mathbf{u}_S)\|_2 \tag{30}$$

$$\leq k_G \|\mathbf{u}^* - \mathbf{u}_S\|_2 + k_G \sum_{i=1}^{n} U_i \tag{31}$$

$$\leq k_G \delta + k_G \sum_{i=1}^{n} U_i \;, \tag{32}$$

which concludes the proof. $\qquad\square$

# F Discussions

In this section, we discuss the limitations and potential negative societal impact of this work.

## F.1 Limitations

### F.1.1 Possible Limitation in Data Collection

In this work, we utilize CARLA to generate LiDAR point clouds instead of collecting data in the real world. Although the delicacy and realism of simulation technologies are now remarkably close to the real world, there still exists a gap. For example, LiDAR cannot detect transparent objects, and LiDAR generates noise when encountering walls; these characteristics are not well-represented by the simulator. In addition, for each placement, we collected 13,600 frames of data; however, real-world applications would require a significantly larger dataset.

### F.1.2 Possible Limitation in LiDAR Configurations

Although we selected seven simplified placements as baselines for extensive experimentation, our baseline placements cannot reflect and cover all possible LiDAR arrangements for various car manufacturers. Furthermore, the optimized placements we obtained are only near-optimal for our dataset. Identifying the globally optimal placements requires further analysis. In addition, our dataset employs spinning LiDAR technology. Yet, the latest advancements in autonomous driving have indicated a trend toward adopting semi-solid-state and solid-state LiDAR systems, necessitating further research. To demonstrate the impact of LiDAR placement, we selected LiDAR sensors with relatively low resolution. The outcomes might differ when the total number of LiDAR sensors is not four or when using LiDAR sensors with different channels and scanning frequencies.

### F.1.3 Possible Limitation in Surrogate Metric

When generating the SOG (semantic occupancy grids) for computing the M-SOG metric, we use a voting algorithm to determine the semantic label for each voxel. This approach can introduce potential errors in cases where multiple semantic labels within a voxel have a similar number of points. Additionally, to accurately describe the semantic distribution of a scene, smaller voxel sizes are often required.

## F.2 Potential Societal Impact

LiDAR systems, by their very nature, are designed to capture detailed information about the environment. This can include not just the shape and location of objects, but potentially also capturing point cloud data of individuals, vehicles, and private property in high resolution. The data captured might be misused if not properly regulated and secured. Further, With autonomous vehicles navigating spaces using LiDAR and other sensors, there could be a shift in how spaces are designed, potentially prioritizing efficiency for autonomous vehicles over human-centered design principles. In addition, over-reliance on LiDAR and perception systems can lead to questions about the trustworthiness and reliability of these systems.

# G  Public Resources Used

We acknowledge the use of the following public resources, during the course of this work:

- CARLA[2] ............................................................ MIT License
- Scenario Runner[3] .................................................. MIT License
- MMCV[4] ......................................................... Apache License 2.0
- MMDetection[5] .................................................. Apache License 2.0
- MMDetection3D[6] ............................................... Apache License 2.0
- MMEngine[7] ..................................................... Apache License 2.0
- nuScenes-devkit[8] ............................................... Apache License 2.0
- SemanticKITTI-API[9] ................................................ MIT License
- MinkowskiEngine[10] ................................................. MIT License
- TorchSparse[11] ..................................................... MIT License
- SPVNAS[12] ......................................................... MIT License
- PolarSeg[13] .............................................. BSD 3-Clause License
- Cylinder3D[14] .................................................. Apache License 2.0
- PointPillars[15] ..................................................... MIT License
- CenterPoint[16] ..................................................... MIT License
- BEVFusion[17] ................................................... Apache License 2.0
- FSTR[18] ......................................................... Apache License 2.0
- Robo3D[19] ................................................... CC BY-NC-SA 4.0

[2] https://github.com/carla-simulator/carla.
[3] https://github.com/carla-simulator/scenario_runner.
[4] https://github.com/open-mmlab/mmcv.
[5] https://github.com/open-mmlab/mmdetection.
[6] https://github.com/open-mmlab/mmdetection3d.
[7] https://github.com/open-mmlab/mmengine.
[8] https://github.com/nutonomy/nuscenes-devkit.
[9] https://github.com/PRBonn/semantic-kitti-api.
[10] https://github.com/NVIDIA/MinkowskiEngine.
[11] https://github.com/mit-han-lab/torchsparse.
[12] https://github.com/mit-han-lab/spvnas.
[13] https://github.com/edwardzhou130/PolarSeg.
[14] https://github.com/xinge008/Cylinder3D.
[15] https://github.com/nutonomy/second.pytorch.
[16] https://github.com/tianweiy/CenterPoint.
[17] https://github.com/mit-han-lab/bevfusion.
[18] https://github.com/Poley97/FSTR.
[19] https://github.com/ldkong1205/Robo3D.

