# OpenReview forum: "Is Your LiDAR Placement Optimized for 3D Scene Understanding?"
_NeurIPS.cc/2024/Conference — NeurIPS 2024 spotlight_

### Official Review · Reviewer_iXBb · 2024-07-10

**Soundness:** 3
**Presentation:** 3
**Contribution:** 3
**Rating:** 6
**Confidence:** 2

**Summary:**

This paper proposes Place3D, a full-cycle pipeline that encompasses LiDAR placement optimization, data generation, and downstream evaluations. The framework makes three appealing contributions. 1) To identify the most effective configurations for multi-LiDAR systems, this paper introduces a Surrogate Metric of the Semantic Occupancy Grids (M-SOG) to evaluate LiDAR placement quality. 2) Leveraging the M-SOG metric, this paper propose a novel optimization strategy to refine multi-LiDAR placements. 3) Centered around the theme of multi-condition multi LiDAR perception, the authors collect a 364,000-frame dataset from both clean and adverse conditions. Extensive experiments demonstrate that LiDAR placements optimized using palce3D outperform various baselines.

**Strengths:**

- This paper first attempt at investigating the impact of multi-LiDAR placements for 3D semantic scene, which is seldom explored in LiDAR-based detection and segmenation areas.
- This paper proposes a more comprehensive metric M-SOG. Experiments show M-SOG is more relevant to the model performance than S-MIG, demonstrating its effectiveness to represent the scene coverage.
- Guided by M-SOG, a LiDAR placement optimization methods is proposed in this work which can provide a solid baseline for LiDAR placement optimization.

**Weaknesses:**

- The representation may be a little confused. The M-SOG and the optimization method, under my understand, is task-independent. However, there are two optimized LiDAR placement related to Det and Seg tasks, which is a little confused.
- To demonstrate the effectiveness of M-SOG, it's recommended to draw a scatter plot of M-SOG and model performance in Tab2 and Tab3, which can be more intuitive.

**Questions:**

- I'm curious about the influence of the LiDAR numbers. It's better to add a experiment to explore the most cost-effective LiDAR number.
- M-SOG seems to be a global metric. Can It also reflect the scene coverage of a specific local area?

**Limitations:**

Nan

---

> ### Author Rebuttal · Authors · 2024-08-06
>
> **Dear Reviewer `iXBb`,**
>
> Thanks for devoting time to this review and providing valuable comments.
>
> ---
> >Q: The representation may be a little confusing. The M-SOG and the optimization method, under my understanding, are task-independent. However, there are two optimized LiDAR placements related to Det and Seg tasks, which is a little confusing.
>
> A: Sorry for the confusion and thanks for the comment. To clarify, the M-SOG and placement optimization strategy are task-independent, but the optimization result depends on the semantic classes involved.
>
> For object detection (Det) and semantic segmentation (Seg), the semantic classes differ. Det considers three object categories: Car, Pedestrian, and Bicycle, while Seg includes 21 object and environment classes, such as Building and Fence. Consequently, the placement optimization result varies between Det and Seg due to the different semantic class definitions.
>
> >Q: To demonstrate the effectiveness of M-SOG, it's recommended to draw a scatter plot of M-SOG and model performance in Tab 2 and Tab 3, which can be more intuitive.
>
> A: Thanks for your suggestion. We include the plot in the attached single-page rebuttal PDF.
>
> As illustrated, the results demonstrate a clear correlation, where the performance generally improves for both tasks as the M-SOG increases. While there might be fluctuations in some placements with specific algorithms, the overall relationship follows a linear correlation, highlighting the effectiveness of our M-SOG for computationally efficient sensor optimization purposes.
>
> >Q: I am curious about the influence of LiDAR numbers. It is better to add an experiment to explore the most cost-effective LiDAR number.
>
> A: Thanks for your suggestion. We appreciate the reviewer's comment highlighting the importance of determining the most cost-effective number of LiDARs. In our study, we investigate the strategic placement of 4 LiDARs, as preliminary experiments indicated that a perception system utilizing 4 LiDARs is more cost-effective compared to other configurations. We test the detection and segmentation performance using 1 to 5 LiDARs, respectively. The results of experiments are as follows:
>
> |Method|1x LiDAR|2x LiDAR|3x LiDAR|4x LiDAR|5x LiDAR|
> |-|:-:|:-:|:-:|:-:|:-:|
> |BEVFusion-L|35.1|39.4|46.2|52.5|55.3|
> |Improvement|-|+4.3|+6.8|+6.3|+2.8|
>
> We observe that the detection performance of BEVFusion-L improved as the number of LiDARs increased. The most cost-effective LiDAR number can be 4 LiDARs as 5 LiDARs only have slight improvement. We will include more experiments on this point using other detectors in the revision.
>
> Additionally, we also supplement the segmentation result of SPVCNN using 1 to 5 LiDARs, respectively.
>
> |Method|1x LiDAR|2x LiDAR|3x LiDAR|4x LiDAR|5x LiDAR|
> |-|:-:|:-:|:-:|:-:|:-:|
> |SPVCNN|40.3|56.6|62.7|68.6|69.1|
> |Improvement|-|+16.3|+6.1|+5.9|+0.5|
>
> Similar to detection, we observe improved segmentation performance when the use of more LiDARs. Again, using 4 LiDARs tends to achieve the optimal trade-off between the segmentation accuracy and computational cost. To further consolidate this finding, more results on other LiDAR segmentation methods will be supplemented during this revision.
>
> >Q: M-SOG seems to be a global metric. Can it also reflect the scene coverage of a specific local area?
>
> A: Thanks for your question. Yes, the M-SOG can be adapted to reflect the scene coverage of specific local areas by segmenting the scene into smaller regions.
>
> The score of M-SOG is connected to the voxels traversed by LiDAR rays in the scene. In our study, the LiDAR is set to sweep [-24.6, 2.0] vertically and [-180.0, 180.0] horizontally in degree. To compute the M-SOG of local areas, we can reset the sweep range of LiDAR rays according to the specific local areas we are interested in, and then get the local traversed voxels. By performing the same process of global M-SOG to local traversed voxels, we can obtain the M-SOG scores which reflect the scene coverage of specific areas.

---

> > ### Author Response · Authors · 2024-08-10
> > **Looking Forward to Discussion with You**
> >
> > **Dear Reviewer `iXBb`,**
> >
> > We sincerely appreciate your time and effort in reviewing our manuscript and providing valuable feedback.
> >
> > ---
> >
> > In response to your insightful comments, we have made the following revisions to our manuscript:
> >
> > - We have clarified the task-independent nature of M-SOG and the optimization method and explained the variations in optimized placements for different tasks.
> > - We have added experiments to explore the influence of the number of LiDARs on performance, showing the most cost-effective LiDAR numbers.
> > - We have included discussions on the local area coverage using M-SOG, adapting the metric to reflect specific regions.
> > - We have added scatter plots of the M-SOG and model performance, which can be found in the attached single-page Rebuttal PDF file.
> >
> > ---
> >
> > We hope these revisions adequately address your concerns.
> >
> > We look forward to actively participating in the **Author-Reviewer Discussion** session and welcome any additional feedback you might have on the manuscript or the changes we have made.
> >
> > ---
> >
> > Once again, we sincerely thank you for your contributions to improving the quality of this work.
> >
> > Best regards,
> >
> > The Authors of Submission 1401

---

### Official Review · Reviewer_PfQe · 2024-07-12

**Soundness:** 3
**Presentation:** 3
**Contribution:** 4
**Rating:** 7
**Confidence:** 5

**Summary:**

The paper targets an overlooked but important aspect of 3D scene understanding -- the influence of LiDAR sensor placements for the success of 3D perception tasks, such as 3D detection and 3D segmentation.

The authors introduce Place3D, a comprehensive pipeline for optimizing LiDAR placement in autonomous driving scenarios to enhance 3D scene understanding, especially under some adverse conditions.

Indeed, current single-LiDAR systems and datasets tend to fail to capture the complexities of real-world environments. This work proposes to address this by using the Surrogate Metric of the Semantic Occupancy Grids (M-SOG) to evaluate LiDAR placement quality. M-SOG measures the information gain and scene understanding capability by calculating the entropy of semantic voxel distributions.

The optimization strategy employs the Covariance Matrix Adaptation Evolution Strategy (CMA-ES) to iteratively improve LiDAR configurations. The objective function is the M-SOG score, ensuring that the optimized placements are close to the global optimum. Compared to the existing S-MIG, this work makes a significant contribution by extending the binomial occupancy distribution to multinomial occupancy distribution through multiple semantic labels for better scene understanding.

Place3D also introduces a dataset of 364,000 frames collected under various conditions, including severe weather and sensor failures. The optimized placements show superior performance in LiDAR semantic segmentation and 3D object detection tasks compared to existing baselines.

**Strengths:**

1. This work focuses on a less-explored but rather important perspective of 3D scene understanding. The established LiDAR placement evaluation benchmark and the proposed placement optimization method could open up some new and interesting research directions in related research areas.

2. An innovative metric,  M-SOG, for evaluating LiDAR placement quality by incorporating semantic information, which improves accuracy over previous methods. Additionally, this work contributes a full-cycle approach from LiDAR placement optimization to data generation and downstream evaluation.

3. The proposed Place3D dataset includes several adverse conditions for realistic and practical evaluation, demonstrating significant improvements in robustness. Additionally, the use of CMA-ES ensures that LiDAR placements are close to the global optimum, with strong theoretical optimality certification.

**Weaknesses:**

1. In the experiments, although the baseline LiDAR placement methods are based on existing configuration of current autonomous vehicle systems (Fig. 7), it is better to compare the proposed method with some randomly sampled LiDAR placements. This can further justify the effectiveness of CMA-ES compared to a wider range of scenarios.

2. It is not clear how the density of the occupancy grid (Sec. 3.3) will influence the surrogate metric, as the estimation of semantic distribution depends on the number of samples.

3. The experiments are conducted in the simulation. Although CARLA is a rather realistic simulator and has been widely adopted in academia, it is unknown how well the proposed method works in the real-world setting. Perhaps it is not feasible to conduct on-site experiments, but more justifications regarding this aspect are needed for further discussion.

4. The optimization process and data generation might require substantial computational resources. Either the main body or the appendix omitted such details. It is highly recommended to supplement such details as they are critical for follow-up works.

5. Can the proposed benchmarks and optimization strategies be extended to other 3D scene understanding tasks, such as 3D object tracking, semantic occupancy prediction, etc?

**Questions:**

Please refer to the above section for detailed questions.

**Limitations:**

The authors have included some discussions of the potential limitations and societal impact in the appendix.

---

> ### Author Rebuttal · Authors · 2024-08-06
>
> **Dear Reviewer `PfQe`,**
>
> Thanks for devoting time to this review and providing valuable comments.
>
> ---
>
> >Q: In the experiments, although the baseline LiDAR placement methods are based on the existing configuration of current autonomous vehicle systems (Fig 7), it is better to compare the proposed method with some randomly sampled LiDAR placements. This can further justify the effectiveness of CMA-ES compared to a wider range of scenarios.
>
> A: Thanks for the insightful comments. We acknowledge that the theoretical upper bound for the performance of the placements optimized by our method cannot be exceeded by random LiDAR placements.
>
> Although enumerating all possible LiDAR placements for random selection is computationally infeasible, we add experiments on evaluating the performance of three random placement strategies as follows:
> |Method|Center|Line|Pyramid|Square|Trapezoid|Line-roll|Pyramid-roll|Random 1|Random 2|Random 3|Ours|
> |-|:-:|:-:|:-:|:-:|:-:|:-:|:-:|:-:|:-:|:-:|:-:|
> |PolarNet|71.0|67.7|67.7|69.3|66.8|67.2|70.9| 65.4 | 64.7 | 65.1 | 76.7
> |MinkUNet|65.7|59.7|62.7|60.7|59.0|58.5|62.2| 60.1 | 59.6 | 59.9 | 66.5
> |SPVCNN|67.1|59.3|67.6|63.4|61.0|60.6|67.9| 59.4 | 59.1 | 59.6 | 68.3
> |Cylinder3D|72.7|68.9|68.4|69.9|68.5|69.8|69.3| 68.0 | 66.7 | 67.9 | 73.0
>
> As can be seen from the above study, randomly sampled placements tend to be sub-par compared to the existing and our optimized strategies. This verifies the effectiveness of the proposed CMA-ES under more general scenarios.
>
> >Q: It is not clear how the density of the occupancy grid (Sec 3.3) will influence the surrogate metric, as the estimation of semantic distribution depends on the number of samples.
>
> A: Thanks for your question.
>
> 1) Density of samples: As the number of samples used to merge and synthesize dense semantic occupancy grids increases, the accuracy of M-SOG metrics is enhanced. This improvement is attributed to the smoother and more continuous semantic representation of the scene resulting from the increased density of semantic LiDAR points. A higher sample density provides a more detailed and granular depiction of the environment, reducing gaps and ambiguities.
>
> 2) Density of voxels: As the grids become denser, the M-SOG scores for all placements will slightly increase, and the linear consistency between our metric and perception performance improves. However, with increasing density, linear consistency of M-SOG may converge as the voxel size decreases to very small dimensions. In the experiments, we used voxel sizes matching those in the detection or segmentation training parameters to compute M-SOG, aiming for better linear consistency. For example, for segmentation, we used voxel sizes of [0.10, 0.10, 0.10]. We computed the M-SOG of other voxel sizes for comparison as follows:
> |Voxel Grid Size|Line-roll|Trapezoid|Line|Pyramid|Square|Pyramid-roll|Center|Ours|
> |-|:-:|:-:|:-:|:-:|:-:|:-:|:-:|:-:|
> |[0.20, 0.20, 0.20]|-5.64|-6.72|-6.13|-5.88|-4.30|-4.24|-4.52|-4.70|
> |[0.10, 0.15, 0.20]|-3.80|-3.65|-3.22|-3.29|-2.67|-2.39|-2.47|-2.28|
> |[0.10, 0.10, 0.10]|-3.13|-2.89|-2.62|-2.56|-2.35|-1.63|-1.58|-1.29|
> |[0.05, 0.10, 0.10]|-2.97|-2.75|-2.40|-2.33|-2.12|-1.37|-1.24|-1.13|
>
> >Q: The experiments are conducted in the simulation. Although CARLA is a rather realistic simulator and has been widely adopted in academia, it is unknown how well the proposed method works in a real-world setting. Perhaps it is not feasible to conduct on-site experiments, but more justifications regarding this aspect are needed for further discussion.
>
> A: Thanks for your insightful comments. We appreciate the reviewer's concern regarding the applicability of our proposed method in real-world settings.
>
> The main challenge of conducting real-world experiments is the inability to ensure identical scenarios for testing different placements. Given multiple LiDAR placements, we need to collect data multiple times in the same area. Even if we ensure that the ego vehicle follows the exact same route and the data is collected at the same time each day, we cannot guarantee the activities of traffic participants within the scene will be identical. This variability in scenarios can lead to difficulties in making fair comparisons.
> Real-world experiments are also time-consuming and resource-intensive, which can limit the extent and frequency of testing. We will include more discussion of extensions to real-world settings in the revised version.
>
> >Q: The optimization process and data generation might require substantial computational resources. Either the main body or the appendix omitted such details. It is highly recommended to supplement such details as they are critical for follow-up work.
>
> A: Thanks for your suggestion. For data generation in CARLA, we recommend using at least 8GB GPUs. For the optimization process, the hardware requirements primarily depend on the resolution of the voxel representation used for the scene.
>
> A CPU with a substantial number of cores and at least 64GB of RAM is necessary for efficient optimization through parallel processing.  The amount of RAM needed scales with the resolution of the voxel representation. We will include details in the revision.
>
> >Q: Can the proposed benchmarks and optimization strategies be extended to other 3D scene understanding tasks, such as 3D object tracking, semantic occupancy prediction, etc?
>
> A: Thanks for your question. Yes, the proposed benchmarks and optimization strategies can indeed be extended to other 3D scene understanding tasks, such as 3D object tracking and semantic occupancy prediction, provided we can obtain the semantic prior knowledge from the training dataset.
>
> In fact, the M-SOG is task-agnostic as it only evaluates the overall semantic coverage quality provided by the LiDAR placements. The differences in optimization results only arise from the varying semantic settings defined in different tasks.

---

> > ### Author Response · Authors · 2024-08-10
> > **Looking Forward to Discussion with You**
> >
> > **Dear Reviewer `PfQe`,**
> >
> > We sincerely appreciate your time and effort in reviewing our manuscript and providing valuable feedback.
> >
> > ---
> >
> > In response to your insightful comments, we have made the following revisions to our manuscript:
> >
> > - We have included additional experiments comparing the proposed method with randomly sampled LiDAR placements to justify the effectiveness of our method.
> > - We have included details on the computational resources required for the optimization process and data generation.
> > - We have discussed the feasibility of extending our benchmarks and optimization strategies to other 3D scene understanding tasks.
> > - We have supplemented details on the influence of the density of the occupancy grid on the surrogate metric.
> >
> > ---
> >
> > We hope these revisions adequately address your concerns.
> >
> > We look forward to actively participating in the **Author-Reviewer Discussion** session and welcome any additional feedback you might have on the manuscript or the changes we have made.
> >
> > ---
> >
> > Once again, we sincerely thank you for your contributions to improving the quality of this work.
> >
> > Best regards,
> >
> > The Authors of Submission 1401

---

> > ### Comment · Reviewer_PfQe · 2024-08-10
> >
> > Thanks to the authors for the comprehensive responses to my review comments.
> >
> > The major concerns from my comments have been resolved.
> >
> > I have also gone through the author’s responses to other three reviewers’ comments.
> >
> > I am confident that this work is of a good value in the area of research and could open up future research opportunities.
> >
> > Therefore, I decide to change the rating from “Weak Accept” to “Accept”.

---

### Official Review · Reviewer_DpmT · 2024-07-13

**Soundness:** 2
**Presentation:** 2
**Contribution:** 3
**Rating:** 3
**Confidence:** 4

**Summary:**

The authors present a very interesting study of optimal lidar placement on autonomous vehicles. They introduce a novel measure of the quality of lidar placement called M-SOG, and also discuss an optimization approach to find the best placement. They measure the impact of the placement on the important vision tasks in autonomous vehicles, such as object detection and semantic segmentation, showing that the placement metric is aligned with the improved quality. Lastly, in addition to clear weather they also analyze the impact on data when the conditions are degraded (such as fog or faulty sensors). Moreover, they also provide the data and the code that supports their work.

**Strengths:**

- Very interesting and timely work.
- The new metric and the results could be of interest to the self-driving community, and the methods should not be difficult to implement.
- The presented results are quite promising.

**Weaknesses:**

- The paper writing is can be improved significantly, as the method is not actually clearly explained.
- The authors tried to stuff too many things in a single paper (metric and degraded sensors), which degraded focus and impact of the work.
- The experiments can be improved significantly.
(detailed comments on the above weaknesses can be found in the section below)

**Questions:**

- Line 107, "voxelized grips" -> "voxels"?
- Line 110, "where each voxel is assumed to be independently and identically distributed", what does this actually mean? The voxels are clearly not independent of each other.
- The notation can be improved, as it is confusing at times and not properly introduced. E.g., line 112, what is T, 1.
- Along similar lines, H is not properly introduced a few lines below.
- Line 143, should P_SOG also have k-index in its name? As it is defined for a particular k, as per the given equation.
- In the paragraph starting with line 150, the authors discuss the metric. However, given that the metric is computed using only a subset of voxels, is there an invalid setup that the metric would not capture properly, such as putting lidars below the car or something similar?
- Fig 2, would be good to zoom in on the roof of the car to make it more visible, as that is the relevant part in this figure.
- The optimization in Section 3.3 is not well explained, and the notation is also not properly introduced.
- E.g., the authors do not properly explain what is the lidar configuration space, or what are the constraints. Some details can be gleaned here and there, but given how critical this part is for their work the authors should do a much better job explaining it.
- Along these lines, line 173 mentions cov-mat of distribution, distribution of what? This is very poorly explained.
- Also, in this paragraph the authors use k-index for samples, but k was already used for classes?
- How significant and how practical is the theoretical analysis? The authors introduce C_M as a gap, but this constant is not discuss anywhere, or how large it is. This makes the theoretical discussion quite weak. More discussion and more clarifications are needed.
- Line 238, Tables 2 and 3 don't actually show this.
- The detectors used in these experiments are not clearly discussed.
- In general, it seems that the authors tried to stuff too many things at the same time in the work and in the experiments. This made them relegate many results and discussions to the appendix, even those that are critical to the work. It seems that it would be much better to focus on one aspect first (e.g., metric + impact on vision results in clear weather) and only then expand from there.
- What are the points in Fig 4, not clear and not discussed.
- The adverse aspect is very important, yet the authors do not spend a lot of time/space on it, and the results are just given. This is unfortunate, and weakens their argument around this part of their work.
- Line 253, this was already mentioned in line 238?
- Line 263, why is the placement good for adverse weather? Any deeper insights? How generalizable are these placements? All these important questions are not discussed at all, and the authors just skip them. This is another symptom of stuffing too many things in one paper and losing focus and depth.
- Line 270, the authors just move all important ablation results to the appendix, which is not acceptable. At this point the authors are just abusing the appendix. Please note that the appendix is not meant to increase the page limit, and critical experiments/results/discussions should NOT be moved there.
- More discussions and insights about the found placement should be added, such as around 2D or roll solution. Any advice for the designers, any interesting findings? This is not provided at all.

###################
FEEDBACK AFTER THE REBUTTAL
###################

I would like to thank the authors for responding to my comments, and also to the other reviewers who helped me further understand the paper. However, after reading through all the other responses/discussion it seems that my main concerns were not addressed. E.g.,while I said that the paper is too stuffed and poorly explained, the authors just said that they will add more explanations and that's that. First, I'm not sure where would this be added since there is no space and the paper is overstretched as is, and second, the required changes are significant and just saying that it will be addressed is not enough, the paper would need to be re-reviewed. Moreover, the authors marked my comments under "Other Minor Modifications", which just shows that they did not appreciate the severity of my comments.
I do like the idea and I do think that the work has future, but with the current execution I just can not see how can I increase my recommendation. The paper needs quite a lot of attention and rewriting, so much so that I am not confident that the authors would not be able to do that with quick updates, especially given their responses that made light of some of my more important comments.

**Limitations:**

The authors do not discuss the limitations in a separate section.

---

> ### Author Rebuttal · Authors · 2024-08-06
>
> **Dear Reviewer `DpmT`,**
>
> Thanks for devoting time to this review and providing valuable comments.
>
> ---
> >Q: Writing can be improved.
>
> A: Thanks for your comment. Due to limited space for response, for minor modifications (e.g. notations), please refer to the above Author Rebuttal section. For specific questions, please refer to the following.
> >Q: The voxels are not independent of each other.
>
> A: Sorry for the confusion. The voxels in a 3D space are not independent of each other due to spatial correlations in a single frame. We intended to convey that the probability distribution of occupancy for each voxel can be modeled independently across a large number of frames.
> >Q: Is there an invalid setup where the metric would not capture properly?
>
> A: Thanks for asking. In our optimization, we consider LiDAR placement within a limited hyper-rectangle space on the vehicle roof. This ensures practical and relevant placements, avoiding physically or operationally infeasible configurations.
> >Q: The optimization in Sec 3.3 is not well explained.
>
> A: In Sec 3.3, we introduce an optimization approach using the Covariance Matrix Adaptation Evolution Strategy (CMA-ES) to find optimal LiDAR placements. This method iteratively searches and refines LiDAR configurations in hyper-rectangle solution space to maximize the M-SOG based objective function. We will include more in the revision.
> >Q: How significant and practical is theoretical analysis?
>
> A: Thanks for the question. The theoretical analysis gives a guarantee for the optimality of the local optima from heuristic CMA-ES under the assumption of the Lipschitz constant $k_G$ and the distance $C_M$ between the maximal and minimal of objective function G. As we have shown on Line 201-204,  Lipschitz constant $k_G$ and the difference between maximum and minimum of G $C_M$  over S can be approximated through calculating $G(u^k_i )$ of Algorithm 1 for each sampled $u^k_i$  over the δ density Grids subset. More specifically, in each iteration $k$, $k_G \sim = \max_i (G(u^k_{i+1}) - G(u^k_i))/(u^k_{i+1} - u^k_i)$, $C_M \sim= \max_i G(u^k_i) - \min_i G(u^k_i)$. As a special case, if input space $U$ is bounded as hyper-rectangle $U$ (as in our experiments), the calculation $C_M$ can be omitted as shown in Corollary 1. We humbly argue that the theoretical analysis is significant in giving theoretical insight into the optimization of NN performance from semantic transformation input (e.g. LiDAR placement in Euclidean space), which may inspire robustness analysis from other semantic transformation perturbations, e.g., rotation, translation, brightness change, etc.
> >Q: Line 238, Tabs 2 and 3 don't show the correlation between M-SOG and performance.
>
> A: Thanks for your comment. Tabs 2 & 3 show the performance of different LiDAR placements, and Tab 1 shows the M-SOG of LiDAR placements. We add a plot of M-SOG and performance in the rebuttal PDF to illustrate it better.
> >Q: The detectors used in these experiments are not clearly discussed.
>
> A: Thanks for your comment. We used 4 detectors including PointPillars, BEV method CenterPoint, and BEVFusion-L, spatio-temporal representation method FSTR, as discussed in Appendix B.3. We will include more discussion in the revision.
> >Q: The authors tried to stuff too many things in one paper.
>
> A: Thanks for your comment. We intend to provide a comprehensive view of our contributions, including the introduction of M-SOG metric, the optimization of LiDAR placements, and the evaluation of placements under both clean and adverse weather conditions. Covering all these aspects is essential to fully demonstrate the value and impact of our method.
> >Q: What are the points in Fig 4?
>
> A: Thanks for asking. Fig 4 illustrates comparisons to existing sensor placement methods, as discussed in Sec 4.2 (Lines 244 to 251). We compare S-MIG [34] and our M-SOG in terms of the linear consistency between metrics and performance, with M-SOG doing better. Greater linear consistency indicates that metrics can effectively predict perception performance, enhancing downstream optimization tasks.
> >Q: The authors do not spend a lot of time/space on the adverse aspect.
>
> A: Thanks for the comment. In Sec 4.1, we detailed the types of adverse conditions considered, including severe weather, external disturbances, and sensor failures. In Sec 4.2, we validated the performance of optimized LiDAR placements under adverse conditions, showing consistent outperformance compared to baselines. In Sec 4.3, we investigated how our method informs LiDAR placements to improve robustness against corruptions. As suggested, we will further explain and analyze the adverse aspects during revision.
> >Q: Why is the placement good for adverse weather?
>
> A: Thanks for asking. The M-SOG utilizes semantic occupancy as prior knowledge, which is invariant to changing conditions, thereby enabling robustness of optimized placement in adverse weather. We also design an ablation study to explore the interplay between the optimization method and perception performance under corruptions in Sec 4.3.
> >Q: How generalizable are these placements?
>
> A: To clarify, our method is data-driven, and the optimized LiDAR placement aligns with the deployment scenarios. This is common industry practice that deploy specific vehicle configurations for specific country or region. However, our method itself is generalizable for any scenario.
> >Q: More discussions and insights about the found placement. Any interesting findings?
>
> A: Our experiments revealed several findings:
> - LiDAR heights: Increasing the average height of the 4 LiDARs improves performance, likely due to an expanded field of view.
> - Variation in heights: A greater difference in LiDAR heights enhances perception, as varied height distributions capture richer features from the sides of objects.
> - Uniform distribution: For Seg, the Pyramid placement performs better, as a more spread-out and uniform distribution of 4 LiDARs captures continuous surface.

---

> > ### Author Response · Authors · 2024-08-10
> > **Looking Forward to Discussion with You**
> >
> > **Dear Reviewer `DpmT`,**
> >
> > We sincerely appreciate your time and effort in reviewing our manuscript and providing valuable feedback.
> >
> > ---
> >
> > In response to your insightful comments, we have made the following revisions to our manuscript:
> >
> > - We have improved the explanation of the optimization approach using the Covariance Matrix Adaptation Evolution Strategy (CMA-ES) and added more details to the theoretical analysis.
> > - we have clarified our experiments for comparison between our M-SOG and existing placement method S-MIG in Fig 4 and provided several findings and insight into LiDAR placement in our experiments.
> > - We have improved the clarity of our figures, such as zooming in on the relevant parts in Fig 2. To better explain Tabs 2 and 3, we have added a scatter plot of M-SOG and model performance, as shown in the attached single-page Rebuttal PDF.
> > - We have clarified notations and improved the overall writing to make the method more understandable. For typos, We corrected "voxelized grid" to "voxels". For notations, we properly introduced notations $T$ and $H$, included the $k$ index in $P_{SOG}$, and revised the $k$ index in optimization to avoid overlap. We also did a thorough check to ensure all notations were properly introduced and clearly explained.
> >
> > ---
> >
> > Additionally, we provide a list of **Summary of Notations** as follows:
> >
> > |Notation|Explanation|
> > |-|-|
> > |$L$|Length of the Region of Interest (ROI)
> > |$W$|Width of the Region of Interest (ROI)
> > |$H$|Height of the Region of Interest (ROI)
> > |$\delta_L$|Resolution of voxelization in the length dimension
> > |$\delta_W$|Resolution of voxelization in the width dimension
> > |$\delta_H$|Resolution of voxelization in the height dimension
> > |$S$|Set of voxels in the Region of Interest (ROI)
> > |$N$|Total number of voxelized grids
> > |$K$|Total number of semantic labels
> > |$v_i$|Voxel $i$
> > |$p(v_i)$|Probability of voxel $v_i$ being occupied
> > |$y_t$|Frame $t$
> > |$T$|Total number of frames
> > |$H(v_i)$|Entropy of voxel $v_i$
> > |$H(v_{i} &#x7c; L)$|Conditional entropy of voxel $v_i$ given LiDAR placement $L$
> > |$H_{POG}$|Total entropy of POG (Probabilistic Occupancy Grids)
> > |$p_{POG}$|Joint probability of all non-zero voxels in the ROI
> > |$p_{POG &#x7c; L}$|Conditional probability of POG given LiDAR placement $L$
> > |$H_{POG &#x7c; L}$|Conditional entropy of POG given LiDAR placement $L$
> > |$\Delta H$|Information gain of 3D scene understanding
> > |$S_{SOG}$|Semantic occupancy grid
> > |$s_{y_k}^{(t)}$|The set of voxels occupied by semantic label $y_k$ at frame $t$
> > |$p(v_i = y_k)$|Probability of voxel $v_i$ being occupied by semantic label $y_k$
> > |$\mathcal{P}_{SOG}$|Conditional joint distribution of SOG given LiDAR placement $L$
> > |$\mathcal{H}_{SOG}$|Entropy of SOG
> > |$L_j$|LiDAR configuration $j$
> > |$\mathcal{H}^{L=L_j}_{SOG}$|Conditional entropy distribution of P-SOG over the voxel set $S &#x7c; L_j$
> > |$\mathbf{M}_{SOG}(L_j)$|Normalized surrogate metric of SOG
> > |$\mathbb{E}$|Expectation operator
> > |$\mathbf{u}_j$|LiDAR configuration $j$ in objective function $F(\mathbf{u}_j)$
> > |$F(\mathbf{u}_j)$|Objective function for LiDAR configuration $j$
> > |$P(\mathbf{u})$|Physical constraint for LiDAR configuration
> > |$G(\mathbf{u})$|Function representing the surrogate metric
> > |$U$|Set of all potential LiDAR configurations
> > |$\mathbf{u}^*$|Optimal LiDAR configuration
> > |$\mathbf{m}^{(k)}$|Mean vector of the distribution at iteration $k$
> > |$\sigma^{(k)}$|Step size of the distribution at iteration $k$
> > |$\mathbf{C}^{(k)}$|Covariance matrix of the distribution at iteration $k$
> > |$\mathcal{N}(\mathbf{m}^{(k)}, (\sigma^{(k)})^2\mathbf{C}^{(k)})$|Normal distribution for sampling ($k$ will be updated to $g$ to avoid overlap with semantic class)
> > |$\delta$|Density for discretizing the configuration space $U$
> > |$\mathbf{u}_i^{(k)}$|Sampled candidate configuration at iteration $k$
> > |$M_k$|Number of best solutions used to update $m^{(k+1)}$
> > |$w_i$|Weights based on solution fitness
> > |$\mathbf{p}_\mathbf{C}^{(k+1)}$|Evolution path accumulating information about the direction of successful steps
> > |$c_\mathbf{C}$|Learning rate for updating the covariance matrix
> > |$p_{\sigma}$|Evolution path for step size adaptation
> > |$c_{\sigma}$|Learning rate for updating the evolution path $p_{\sigma}$
> > |$d_{\sigma}$|Normalization factor for step size adaptation
> > |$E\|N(0, I)\|$|Expected length of a standard normally distributed random vector
> > |$\| \cdot \|$|Euclidean norm
> >
> > ---
> >
> > We hope these revisions adequately address your concerns.
> >
> > We look forward to actively participating in the **Author-Reviewer Discussion** session and welcome any additional feedback you might have on the manuscript or the changes we have made.
> >
> > ---
> >
> > Once again, we sincerely thank you for your contributions to improving the quality of this work.
> >
> > Best regards,
> >
> > The Authors of Submission 1401

---

### Official Review · Reviewer_8eRR · 2024-07-16

**Soundness:** 3
**Presentation:** 3
**Contribution:** 3
**Rating:** 6
**Confidence:** 4

**Summary:**

This paper proposes a framework called Place3D to investigate the placement of multiple LiDAR sensors for semantic segmentation and object detection task under various weather conditions.  Place3D mainly introduce a Surrogate Metric of Semantic Occupancy Grids (M-SOG) to evaluate the perception performance with different sensors placements.  Besides, an optimization approach is applied to refine the LiDAR placement. The method is evaluated on the simulated dataset with CARLA, showing the advantages of the refined placements.

**Strengths:**

- The motivation is interesting and the paper is well-written.
- A new metric and a dataset are proposed for the evaluation of the placement.
- The experiments are extensive, three typical models are evaluated on 7 weather scenarios.
- Comparison with 7 popular placement methods show the advantages of the refined placement with the proposed methods.

**Weaknesses:**

- Although the proposed method  investigates LiDAR placement for more tasks under more weathers, the core formulation is similar to that in [34] as cited in the paper.
- Only one type of 16-beam LiDAR sensor is investigated. Actually, sensors with more laser beams (64 or even 128) are becoming low-cost and popular. It is important to evaluate the effect of the placement as the point cloud data get dense.
- During the companions in Tab.2,3, different refined placements are applied for different tasks. How is the performance if applied a Seg-optimized placement to the Det task? It is impractical to change the placement for different task. Is that possible to find an optimal placement for both tasks?

**Questions:**

- In the dataset comparison part, it is written that no adverse weather for Nuscenes and Waymo.  They may include rainy weather and even more (e.g., Snow).  Although in the appendix part, please also have a check on that.
- How is the solution spaces for the LiDAR placement? Are the displacements along z-axis or even rotation are considered during the optimization?
- Is there any intuitive interpretation about why refined sensor placement are beneficial to different tasks?

**Limitations:**

The authors discussed the possible limitation in data collection.

---

> ### Author Rebuttal · Authors · 2024-08-04
>
> **Dear Reviewer `8eRR`,**
>
> Thanks for devoting time to this review and providing valuable comments.
>
> ---
>
> >Q: Although the proposed method investigates LiDAR placement for more tasks under more weather, the core formulation is similar to that in [34] as cited in the paper.
>
> A: Thank you for the insightful comment. We acknowledge the reviewer's observation regarding the similarities between our core formulation and [34]. However, our method M-SOG addresses critical limitations present in S-MIG [34] and makes significant improvements as follows:
> - While S-MIG primarily focuses on object binomial occupancy, our M-SOG incorporates semantic differentiation to form a multinomial occupancy distribution, providing a richer and more detailed understanding of the scene. This allows our metric to better capture the true nature of the environment.
> - S-MIG constructs occupancy grids with 3D bounding boxes, overlooking the occlusion relationships between objects and the environment. Our M-SOG addresses this critical limitation by leveraging semantic distribution, which only incorporates the reachable surface of objects and environments for LiDAR rays.
> - We compare S-MIG and our M-SOG in terms of the linear consistency between metrics and perception performance (Lines 244 to 251) and illustrate our superiority in Fig 4. Greater linear consistency enables the surrogate metric to better predict perception performance. Thus M-SOG enhances applicability to downstream optimization tasks.
>
> > Q: Only one type of 16-beam LiDAR sensor is investigated. Sensors with more laser beams (64 or even 128) are becoming low-cost and popular. It is important to evaluate the effect of placement as the point cloud data gets dense.
>
> A: Thanks for your insightful comment. We appreciate the reviewer's comment on the using denser beam LiDARs. In our work, we specifically chose to utilize 16-beam LiDARs to perform experiments, totaling 64 beams with 4 LiDARs, based on the following goals:
> - Using low-resolution LiDARs increases the challenge in both object detection and semantic segmentation, making it easier to observe the impact of LiDAR placements on perception.
> - LiDAR point clouds become sparse at long distances and the use of multiple LiDARs is specifically to enhance the perception of sparse point clouds. To this end, we use low-resolution LiDARs to simulate the conditions of sparse point clouds from high-resolution LiDARs at long distances. This highlights the role of multiple LiDARs in improving the perception of sparse point clouds, especially for distant objects.
>
> We added experiments on 64-beam LiDARs and found that placement remains important with higher-resolution LiDARs. The results are as follows.
> |Beam|Method|Center|Line|Pyramid|Trapezoid|
> |-|-|:-:|:-:|:-:|:-:|
> |16|BEVFusion-L|52.5|49.3|51.0|50.2
> |64|BEVFusion-L|77.1|80.4|79.3|77.5
>
> > Q: During the companions in Tabs 2 and 3, different refined placements are applied for different tasks. How is the performance if applied a Seg-optimized placement to the Det task? It is impractical to change the placement for different tasks. Is it possible to find an optimal placement for both tasks?
>
> A: Thanks for asking. We show the performance of applying Seg-optimized placement to Det tasks and list the performance of Det-optimized layouts as a comparison. While the performance is close, the Seg-optimized placement performance still lags slightly behind the Det-optimized placement performance.
>
> To find optimal placement for both tasks, a possible solution is to define a weighted objective function for optimizing, such as $F(L_j) = M_{SOG|Seg}(L_j) + \lambda M_{SOG|Det}(L_j)$, where $\lambda$ is the weight which should be designed carefully.
>
> |Method|Center|Line|Pyramid|Square|Trapezoid|Line-roll|Pyramid-roll|Seg-optimized|Det-optimized|
> |-|:-:|:-:|:-:|:-:|:-:|:-:|:-:|:-:|:-:|
> |BEVFusion-L|52.5|49.3|51.0|49.2|50.2|50.8|50.7|51.8|53.0
> |CenterPoint|55.8|54.0|55.9|54.0|56.4|55.2|56.2|56.4|57.1
>
> >Q: In the dataset comparison part, it is written that there is no adverse weather for nuScenes and Waymo. They may include rainy weather and even more (e.g., Snow). Although in the appendix part, please also have a check on that.
>
> A: Sorry for the oversight and thanks for your comment. We will address this in the revision.
>
> While nuScenes and Waymo include adverse weather, our dataset provides categorized weather data under identical scenarios. This improvement allows for systematic study and fair evaluation of the degradation of perception models in adverse conditions, compared to the naturally distributed weather data in nuScenes and Waymo.
> >Q: How are the solution spaces for the LiDAR placement? Are the displacements along the Z-axis or even rotation considered during the optimization?
>
> A: Thanks for asking. In our work, Z-axis and rotation are exactly considered in the optimization. The solution space for LiDAR placement in our optimization framework is defined as a 3D cuboid space above the vehicle roof. This space includes both the location coordinates (X, Y, Z) and roll angles for each LiDAR unit. Specific placements and a detailed explanation are discussed in Appendix B.1.
> >Q: Is there any intuitive interpretation of why refined sensor placement is beneficial to different tasks?
>
> A: Thanks for your question. There are several intuitive reasons why refined sensor placement is beneficial.
> - LiDAR heights: Increasing the average height of the 4 LiDARs improves performance, likely due to an expanded field of view.
> - Variation in heights: A greater difference in LiDAR heights enhances perception, as varied height distributions capture richer features from the sides of objects.
> - Uniform distribution: The pyramid placement performs better in segmentation, as a more spread-out and uniform distribution of 4 LiDARs captures richer surface features.
>
> As discussed, our optimized placements are anticipated to align with these findings in the experiments, leading to better performance.

---

> ### Author Response · Authors · 2024-08-10
> **Looking Forward to Discussion with You**
>
> **Dear Reviewer `8eRR`,**
>
> We sincerely appreciate your time and effort in reviewing our manuscript and providing valuable feedback.
>
> ---
>
> In response to your insightful comments, we have made the following revisions to our manuscript:
>
> - We have clarified the core formulation and highlighted the improvements over the work cited in [34], specifically addressing critical limitations in S-MIG.
> - We have expanded our experiments to include higher-resolution LiDARs, demonstrating that placement remains important even with denser point clouds.
> - We have provided intuitive explanations of how refined sensor placements enhance performance in various tasks, such as improving the field of view and capturing more detailed features.
> - We have verified the presence of adverse weather conditions in the Waymo and nuScenes datasets and detailed the key differences between our dataset and others in terms of adverse weather scenarios.
>
> ---
>
> We hope these revisions adequately address your concerns.
>
> We look forward to actively participating in the **Author-Reviewer Discussion** session and welcome any additional feedback you might have on the manuscript or the changes we have made.
>
> ---
>
> Once again, we sincerely thank you for your contributions to improving the quality of this work.
>
> Best regards,
>
> The Authors of Submission 1401

---

### Author Rebuttal · Authors · 2024-08-06

**Dear Reviewers, Area Chairs, and Program Chairs,**

We sincerely thank the reviewers, ACs, and PCs for the time and efforts devoted during this review.

We especially appreciate our reviewers for offering valuable comments, providing positive feedback, and drawing insightful suggestions.

---

We are encouraged that our reviewers recognize this work:
- **Reviewer `8eRR`:**
  - *"has interesting motivation and extensive experiments"*, *"proposes a new metric and a dataset"*, and *"the paper is well-written"*.
- **Reviewer `DpmT`:**
  - *"is very interesting and timely"*, *"could be of interest to the self-driving community"*, and *"results are quite promising"*.
- **Reviewer `PfQe`:**
  - *"focuses on a less-explored but rather important task"*, "*could open up new and interesting research directions in related research areas"*, *"contributes a full-cycle approach from LiDAR placement optimization to data generation and downstream evaluation"*, and "*has strong theoretical optimality certification"*.
- **Reviewer `iXBb`:**
  - *"presents the first attempt at investigating the impact of multi-LiDAR placements for the 3D semantic scene"*, *"demonstrates effectiveness to represent the scene coverage"*, and "*provides a solid baseline for LiDAR placement optimization"*.

---

As suggested by our reviewers, we have revised the manuscript accordingly. We present a **summary of changes** as follows:

- **Methods & Technical Details:**
  - As suggested by **Reviewer `8eRR`**, we have clarified the core formulation and highlighted the improvements over the work cited in [34], specifically addressing critical limitations in S-MIG.
  - As suggested by **Reviewer `DpmT`**, we have improved the explanation of the optimization approach using the Covariance Matrix Adaptation Evolution Strategy (CMA-ES) and added more details to the theoretical analysis.
  - As suggested by **Reviewer `PfQe`**, we have included additional experiments comparing the proposed method with randomly sampled LiDAR placements to justify the effectiveness of our method.
  - As suggested by **Reviewer `iXBb`**, we have clarified the task-independent nature of M-SOG and the optimization method and explained the variations in optimized placements for different tasks.

- **Experiments:**
  - As suggested by **Reviewer `8eRR`**, we have expanded our experiments to include higher-resolution LiDARs, demonstrating that placement remains important even with denser point clouds.
  - As suggested by **Reviewer `DpmT`**, we have clarified our experiments for comparison between our M-SOG and existing placement method S-MIG in Fig 4 and provided several findings and insight of LiDAR placement in our experiments..
  - As suggested by **Reviewer `PfQe`**, we have included details on the computational resources required for the optimization process and data generation.
  - As suggested by **Reviewer `iXBb`**, we have added experiments to explore the influence of the number of LiDARs on performance, showing the most cost-effective LiDAR numbers.

- **Elaboration:**
  - As suggested by **Reviewer `8eRR`**, we have intuitive interpretations of why refined sensor placements are beneficial to different tasks, such as increasing the field of view and capturing richer features.
  - As suggested by **Reviewer `DpmT`**, we have improved the clarity of our figures, such as zooming in on the relevant parts in Fig 2. To better explain Tabs 2 and 3, we have added a scatter plot of M-SOG and model performance, as shown in Rebuttal PDF.
  - As suggested by **Reviewer `PfQe`**, we have discussed the feasibility of extending our benchmarks and optimization strategies to other 3D scene understanding tasks.
  - As suggested by **Reviewer `iXBb`**, we have included discussions on the local area coverage using M-SOG, adapting the metric to reflect specific regions.

- **Other Minor Modifications:**
  - As suggested by **Reviewer `8eRR`**, we have checked and verified the presence of adverse weather conditions in Waymo and nuScenes, and included the major difference between our dataset and other datasets in the aspect of adverse weather in our revision.
  - As suggested by **Reviewer `DpmT`**, we have clarified notations and improved the overall writing to make the method more understandable. For typos, We correct "voxelized grid" to "voxels". For notations, we properly introduce notation $T$ and $H$, include the $k$ index in $P_{SOG}$, and revise the $k$ index in optimization to avoid overlap. We also do a thorough check to ensure all notations are properly introduced and clearly explained.
  - As suggested by **Reviewer `PfQe`**, we have supplemented details on the influence of the density of the occupancy grid on the surrogate metric.
  - As suggested by **Reviewer `iXBb`**, we have added scatter plots of the M-SOG and model performance, which can be found in the attached single-page Rebuttal PDF file.

For detailed responses regarding each of the above aspects, please kindly refer to the response **rebuttal windows** in the review section.

---

Last but not least, we sincerely thank our reviewers, ACs, and PCs again for the valuable time and efforts devoted and the constructive suggestions provided during this review.

---

Yours sincerely,

The Authors of Submission 1401

---

### Author Response · Authors · 2024-08-14
**Summary of Author-Reviewer Discussion Session**

**Dear Reviewers, Area Chairs, and Program Chairs,**

As the Author-Reviewer Discussion session comes to a close, we would like to express our sincere gratitude for your time, effort, and thoughtful feedback throughout this review process.

We are pleased that our rebuttal and revisions have addressed many of the reviewers’ concerns, and we appreciate the positive feedback we received.

---

Our revisions were guided by your insightful suggestions, resulting in several **key improvements**:
- We clarified the core formulation and highlighted the improvements over the work cited in [34], specifically addressing critical limitations in S-MIG, as suggested by Reviewer `8eRR`. We also improved the explanation of our optimization approach using CMA-ES and expanded the theoretical analysis as per the Reviewer `DpmT`’s advice.
- We included a scatter plot illustrating the relationship between M-SOG and model performance, making our results more intuitive and easier to interpret, as suggested by Reviewer `iXBb`. Additionally, we clarified our comparisons between M-SOG and S-MIG and added further insights into LiDAR placement based on Reviewer `DpmT`’s suggestions.
- We included higher-resolution LiDAR experiments, demonstrating the continued importance of sensor placement with denser point clouds, as recommended by Reviewer `8eRR`. We also included experiments on the influence of the number of LiDARs, as suggested by Reviewer `iXBb`.
- We provided intuitive interpretations of why refined sensor placements are beneficial for various tasks, such as enhancing the field of view and capturing richer features, as suggested by Reviewer `8eRR`. We also discussed the feasibility of extending our benchmarks and optimization strategies to other 3D scene understanding tasks, following Reviewer `PfQe`’s advice.
- We addressed specific concerns regarding adverse weather conditions in datasets, clarified notations, and improved the overall clarity of our figures and writing. We also included additional experiments and discussions, such as local area coverage using M-SOG, reflecting the valuable input from all reviewers.

---

We would like to highlight again the **technical contributions** of this work:
- To the best of our knowledge, Place3D is the first attempt to investigate the impact of multi-LiDAR placements for 3D semantic scene understanding across diverse conditions. It addresses the limitations of existing datasets that often lack adverse weather and sensor failure scenarios.
- We introduce the Surrogate Metric of Semantic Occupancy Grids (M-SOG), an innovative metric designed to evaluate the quality of LiDAR placements effectively for both detection and segmentation tasks. This metric overcomes the limitations of previous methods by incorporating semantic information and handling occlusions.
- Our novel optimization strategy, utilizing the M-SOG metric, refines LiDAR placements to enhance performance in both LiDAR semantic segmentation and 3D object detection. This approach demonstrates superior robustness and accuracy, outperforming various baselines by approximately 9%.
- We contribute a large-scale, 364,000-frame dataset that includes both clean and adverse conditions, along with a comprehensive benchmark for evaluating LiDAR-based 3D scene understanding. This dataset is crucial for testing the robustness of 3D perception systems under realistic conditions.

---

We hope these revisions have strengthened our manuscript and fully addressed your feedback. We are encouraged by the **positive recognition** of our work’s potential impact, as highlighted by Reviewer `PfQe`’s comments, who has upgraded the rating from *"6: Weak Accept"* to *"7: Accept"*.

---

We sincerely thank you for your constructive suggestions and for helping us improve the quality of this work.

Best regards,

The Authors of Submission 1401

---

### Decision · Program_Chairs · 2024-09-25

**Decision:**

Accept (spotlight)

**Comment:**

The paper presents a metric and optimization strategy to obtain LiDAR placements for autonomous driving that achieve improvements for perception tasks like 3D object detection and semantic segmentation, including under adverse conditions of sensor function and weather. It also develops a dataset and benchmark in an attempt to standardize multi-LiDAR configurations. Reviewers 8eRR, PfQe and iXBb recommend accepting the paper, while DpmT argues for rejection. All the reviewers appreciate the value of the contributions. The main issues raised by DpmT are on improved notations and density of information in the main paper. This was discussed extensively, especially by PfQe and DpmT, where the former champions the paper and finds that the presentation does not occlude the contributions. The AC concurs that the issues raised by DpmT are important, but have been sufficiently addressed by the author rebuttal. The authors are encouraged to include the notational improvements requested by DpmT in the final version of the paper. The paper does solve an under-studied but important problem for autonomous driving, where the proposed dataset and methods will help alleviate significant costs associations with trial-and-error involved in setting up expensive multi-LiDAR systems. In balance, the paper is recommended for acceptance at NeurIPS.